# Diffract: Spectral View of LLM Domain Adaptation

**Nikita Borodin**[1] **Maria Krylova**[1] **Artem Zabolotnyi**[1,2] **Dmitry Aspisov**[1] **Egor Shikov**[1] **Nikita Tyuplyaev**[1]
**Oleg Travkin**[1] **Roman Alferov**[1] **Dmitry Vinichenko**[1]

## Abstract

We study continual pre-training (CPT) as a mechanism for adapting general-purpose large language models to specialized domains: mathematics, instruction, code, and natural text. Using singular value decomposition of weight matrices, we find that CPT leaves singular value spectra largely invariant, with adaptation driven mainly by changes in singular vectors. An analysis of attention-head projection matrices reveals strong, domain-dependent **head heterogeneity**, which we exploit to define a head importance criterion: up to **60%** of head updates can be removed without measurable quality loss. Selectively rewinding low-importance heads to their pre-trained state improves benchmark accuracy by up to **4%** versus the fully trained baseline. Finally, we identify **domain connectivity**—linear interpolation between CPT checkpoints yields smooth domain-quality interpolation without notable degradation on either domain—and release Diffract, an open-source toolkit for scalable spectral analysis of billion-parameter models.

## 1. Introduction

Continual pre-training (CPT) is now a well-established component of modern LLM training pipelines. It is central to producing domain-specialized models, such as DeepSeek-Math (Shao et al., 2024) and Code Llama (Rozière et al., 2023); applying CPT to a general-purpose model yields better performance than training a specialized model from scratch under the same compute budget. CPT is used in many recent state-of-the-art models, such as OLMo 3 (OLMo Team: Ettinger, A. et al., 2026) and Llama 3 (Grattafiori et al., 2024) — the final pre-train stage uses curated, domain-specific data mixtures while the learning

[1]Risk AI Research Lab [2]Applied AI Institute. Correspondence to: Dmitry Vinichenko <dmitry.vinichenko@risk-ai-research.tech>.

*Proceedings of the $43^{rd}$ International Conference on Machine Learning*, Seoul, South Korea. PMLR 306, 2026. Copyright 2026 by the author(s).

rate is linearly annealed to zero. Empirical evidence suggests that this late-stage focus on cleaner, domain-relevant data improves mathematical and coding abilities without degrading general-language performance (Blakeney et al., 2024).

However, the CPT stage of the LLM training pipeline is significantly less studied compared to the supervised fine-tuning (SFT) stage, where a rich set of phenomena has been documented—including linear mode connectivity (Frankle et al., 2020), task arithmetic (Ilharco et al., 2023), model soups (Wortsman et al., 2022), ability transfer (Yu et al., 2024) and low-rank subspace modification (Hu et al., 2022; Meng et al., 2024). To investigate whether similar phenomena exist for CPT, we conduct a series of pre-train and continual pre-training experiments on 1B, 7B, and 13B language models with OLMo 2 architecture for several domains: math, instruction, code, and text. We analyze the CPT delta $\Delta W = W^{\text{domain}} - W^{\text{pre-train}}$, characterize its sparsity, and assess the ability to interpolate between checkpoints adapted to different domains.

Additionally, we investigate the singular spectra of weight matrices for checkpoints along the pre-train trajectory and for weight increments after CPT, and uncover several novel phenomena. Prior random matrix theory-based work suggests that heavy-tailed singular value distributions are closely connected to the generalization ability (Mahoney & Martin, 2019; Martin & Mahoney, 2021; Yang et al., 2023). In our analysis, we consider singular spectra of weight matrices as well as the dynamics of singular vectors, which were earlier shown to play an important role in the model training process (Yunis et al., 2024).

In summary, we highlight our key contributions:

1. Investigation of the dynamics of weight matrix singular value spectra, and identification of the development of complex spectral structure in attention head matrices along the pre-train stage, which cannot be described by heavy-tailed self-regularization theory (Mahoney & Martin, 2019). This is associated with an increase in quality on language tasks and faster domain adaptation on CPT. Moreover, we find that the spectra during CPT remain almost stable, and domain adaptation is

primarily driven by singular vector changes localized near the peaks in the SVD spectra.

2. Identification of **head heterogeneity**, namely, the varying behavior of attention heads during CPT stages on different domains, which becomes more pronounced as the pre-train token budget increases. More specifically, we observe that some heads demonstrate significant changes regardless of CPT domain, while others change in a domain-specific manner. We use this information to put forward a criterion for ordering heads according to their effect on CPT quality, which enables us to achieve a quality increase of up to **4**% for math CPT of the 7B model upon a partial head rewind. Additionally, we discover that CPT deltas have redundancy in parameters, which increases with model size: up to **60**% of attention heads or up to **50**% of the smallest singular values in the CPT delta can be dropped without significant degradation of model quality.

3. Identification of the ability to linearly interpolate between checkpoints after CPT on different domains without quality decrease, for which we coin the term **domain connectivity**; we find that interpolated model quality improves with the increase in pre-train stage token budget.

4. Implementation of an open-source tool for matrix spectra analysis—Diffract package—in order to ensure the reproducibility of our findings and to facilitate future research. The code is available at https://github.com/Risk-AI-Research/diffract

## 2. Methodology

In order to investigate the properties of model weight matrices, we employ several analytical methods centered on singular value decomposition (SVD) (Golub & Reinsch, 1970). For a matrix $W \in \mathbb{R}^{m \times n}$ with $m \geq n$, we write the SVD as

$$W = U \Sigma V^\top,$$

where $U \in \mathbb{R}^{m \times r}$ and $V \in \mathbb{R}^{n \times r}$ have orthonormal columns, $\Sigma = \text{diag}(\sigma_1, \ldots, \sigma_r)$, $r = \text{rank}(W)$, and $\sigma_1 \geq \cdots \geq \sigma_r > 0$. This notation is used consistently throughout our analysis.

**Norms and Ranks.** We characterize singular spectra $\Sigma(W)$ through established spectral measures: the Frobenius norm $\|W\|_F$, spectral norm $\|W\|_2$, stable rank $R^s(W)$, and effective rank $R^e(W)$. Complete definitions are provided in Appendices A.1 and A.2.

**Singular Vector Agreement.** SVD provides both spectral magnitudes and directional information through singular vectors. To analyze directional changes during training, we measure agreement between singular vectors of a given weight matrix along the training trajectory. Further implementation details are presented in Appendix A.3.

**Fitting Model Distributions.** We model the empirical spectral density (ESD) using two complementary approaches: the Marchenko–Pastur distribution (Marčenko & Pastur, 1967) for the bulk spectrum and power law models for heavy-tailed spectral regions. Estimation procedures and diagnostic methods are detailed in Appendix A.4.

## 3. Experimental Setup

### 3.1. Training Setup

We initialize all models using the OLMo 2 architecture and training stack (Walsh et al., 2025), training 1B, 7B, and 13B models. Our experimental pipeline consists of two sequential stages:

**Stage 1: Pre-train.** We pre-train models from scratch on mixtures sampled from DCLM (Li et al., 2024), using token budgets ranging from 20B to 400B. Training follows a cosine learning rate schedule with a warm-up phase; for each pre-train dataset size we use a full scheduling cycle. Due to computational constraints, this stage is conducted only for the 1B model; we also use 1B and 7B checkpoints pre-trained for 4T tokens, 13B checkpoint pre-trained for 5T tokens, and 32B checkpoint pre-trained for 6T tokens provided by OLMo team.

**Stage 2: Continual pre-training (CPT).** Starting from the pre-trained checkpoints listed above, we further train the models on several data mixtures to study domain shift and replay effects. We vary the CPT data composition to emphasize (i) DolminoMath-only data (MATH) (Walsh et al., 2025), (ii) balanced DCLM+DolminoMath mixtures, (iii) DCLM-heavy replay mixtures (TEXT), (iv) instruction data from FLAN (Wei et al., 2022) and Stack Exchange [1] (INST), and (v) instruction data combined with Dolmino-Math and (vi) source code from StarCoder (CODE) (Li et al., 2023). The learning rate is initialized to the final value used in Stage 1 and is annealed to zero throughout this phase. Complete reproducibility details, including hyperparameters, training configurations, and data splits, are provided in Appendix B.2. The code can be found in Appendix B.3.

### 3.2. Evaluation Protocol

The quality of the model is evaluated using the OLMES framework (Gu et al., 2025). Language accuracy is measured as the average performance across three datasets: ARC-Easy (Clark et al., 2018), HellaSwag (Zellers et al., 2019), and WinoGrande (Sakaguchi et al., 2020). Each of these language tasks is evaluated in a 5-shot setting, where

---

[1] https://archive.org/details/stackexchange_20240930

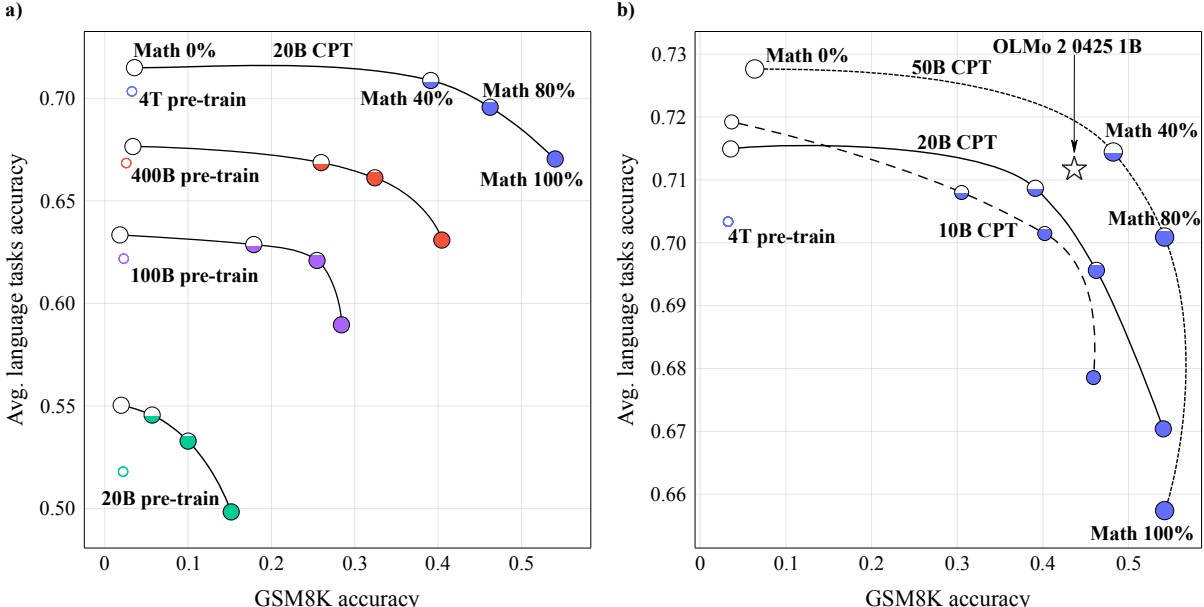

*Figure 1.* Math–language quality trade-off with continual pre-training on the OLMo 2 1B backbone. The y-axis reports the average accuracy on WinoGrande, ARC-Easy, and HellaSwag. Hollow circles denote pre-train checkpoints; filled circles denote continual pre-training runs initialized from the corresponding pre-train checkpoints. Marker size encodes the CPT token budget (10B, 20B, 50B) on a mixture of math and text data; marker fill encodes the math proportion in the data (0%, 40%, 80%, 100%). Points with the same CPT token budget are connected. a) Overview across pre-train sizes (20B, 100B, 400B, 4T). For each pre-train checkpoint, we plot CPT runs with a fixed 20B-token budget (equal marker sizes) and varying math proportions; the leftmost hollow marker in each connected series is the corresponding pre-train checkpoint. b) Quality of CPT runs starting from 4T pre-train checkpoint (leftmost hollow marker). CPT runs vary both the token budget and the math proportion; the star marks the original OLMo 2 0425 1B CPT model (50B tokens with 10B math, i.e., 20%). Larger pre-train token budgets yield better results overall. Increasing the math proportion moves models rightward while typically lowering language-task accuracy, whereas larger token budgets shift the trade-off frontier outward.

answer choices are scored individually using LLM token probabilities in a cloze-style format.

For math accuracy, we use an 8-shot evaluation on GSM8K (Cobbe et al., 2021) and 4-shot evaluation on MATH-500 (Lightman et al., 2024), computing exact-match accuracy between model predictions and the gold answers. To assess instruction following and reading comprehension, we evaluate on the DROP (Dua et al., 2019) and SQuAD (Rajpurkar et al., 2016) datasets. Finally, code generation capabilities are measured using HumanEval (Chen et al., 2021).

## 4. Main Results

### 4.1. CPT Quality Dynamics

First, we focus on identifying trends in the CPT quality as a function of token budget and CPT dataset composition. To that end, we study continual pre-training on the Dolmino-Math mathematics corpus, which demonstrates pronounced quality gains in GSM8K accuracy (Walsh et al., 2025). We run experiments on a 1B model with pre-train token budgets of 20B, 100B, 400B and 4T, and CPT token budgets of 10B, 20B, and 50B for CPT mixtures of DolminoMath and

DCLM data.

Our analysis reveals two key findings. First, we consider CPT runs starting from the 4T pre-train checkpoint, and find that the math performance plateaus at 20B (Fig. 1(b)). GSM8K accuracy for our 20B CPT run is higher than that of the OLMo 2 0425 1B checkpoint. Therefore, in further experiments with other pre-train token budgets, other domains, and the 7B model, we focus on a 20B-token CPT. For the 13B model, however, we use a 10B-token CPT budget, since our additional experiments indicate that this budget is already sufficient to approach the quality close to the released OLMo 2 1124 13B checkpoint.

Second, for 20B CPT runs starting from different pre-train budgets, we find that while math performance after pre-train is similarly low—consistent with the absence of domain knowledge in the pre-train dataset mix—models with a longer pre-train stage achieve better final math quality metrics after the CPT stage (Fig. 1(a)). Similar results hold for 10B and 50B CPT lengths, as we demonstrate in Appendix Fig. 7. In the next section, we propose a hypothesis for such behavior based on spectral analysis of model weight matrices.

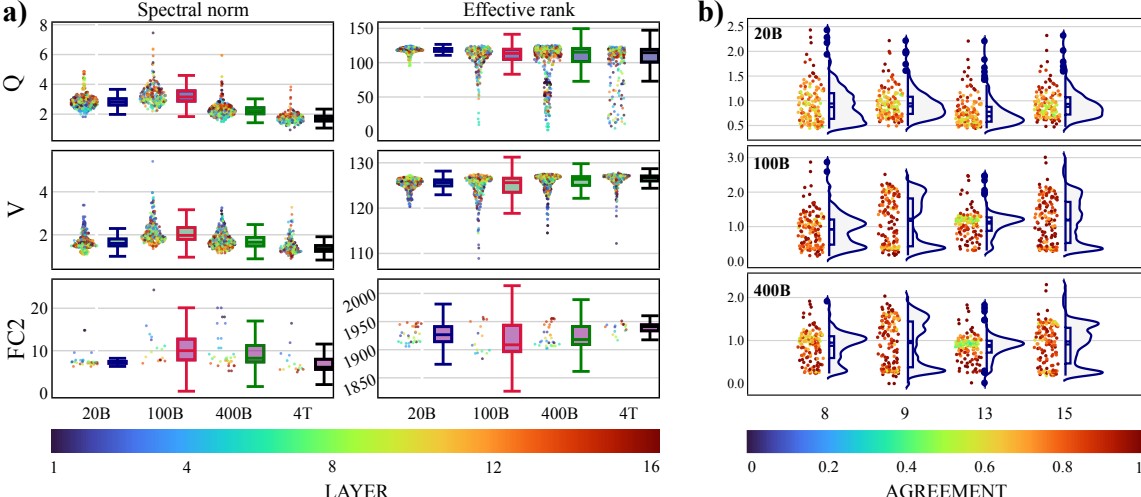

*Figure 2.* Spectral shape and metrics evolution during pre-train. a) Rows (top to bottom) correspond to $W^Q$, $W^V$, and $W^{FC_2}$ weight matrices, while columns (left to right) report the Spectral norm and the Effective rank. Each subplot overlays jittered per-matrix values with box plots summarizing the distributions for models pre-trained on 20B, 100B, 400B, and 4T tokens. Points denote individual matrices and are color-coded by the corresponding layer index. b) Singular value spectra for $W^Q$ (layer 6, heads 8, 9, 13, 15) at 20B, 100B, and 400B tokens. Jitter points represent singular values and are colored by the left singular vector agreement between each pre-train checkpoint and the corresponding CPT endpoint on math domain. Note the increasingly complex spectral shape that poorly conforms to MP and PL models. Starting at 100B the singular vectors that change during the CPT stage are increasingly localized in narrow spectral bands.

### 4.2. Spectral Evolution Along the Pre-Train Stage

To gain insight into pre-train dynamics, we first consider weight matrix norms and ranks (Fig. 2(a)). We highlight the non-monotonic behavior of Frobenius and spectral norms, which reach maximum values at around 100B tokens of pre-train. Effective rank also demonstrates an inflection point around 100B tokens: a substantial number of layer-level outliers with significantly lower rank emerge.

We go beyond basic statistics and perform a detailed examination of the shapes of the singular spectra for attention weights of 1B, 7B, and 13B models. For 1B we consider pre-train budgets from 20B to 400B (Fig. 2(b)). At initialization, weight matrices are random, and their singular spectra follow the Marchenko-Pastur law (Marčenko & Pastur, 1967). After 20B pre-train, the power law tail starts to form in the spectra of the heads (Fig. 2(b), top panel), in accordance with heavy-tailed self-regularization theory (HTSR) (Martin & Mahoney, 2021). However, at 100B and larger pre-train budgets the complexity of attention heads spectra increases—namely, the appearance of outliers and multiple narrow peaks. The same holds for both 7B and 1B models after a 4T pre-train, and for 13B model after a 5T pre-train (Appendix Fig. 8(b)), moreover we observed the same trend on original OLMo 2 checkpoints after CPT for 13B and 32B (Appendix Fig. 9). These empirical distributions deviate significantly from HTSR models. We hypothesize that such complex structure is a prerequisite

for faster domain adaptation of models with larger pre-train token budgets and note that these findings call for the development of more complex models for the singular spectra of attention heads.

The spectral structure of MLP blocks stays close to the HTSR model with power law tail—resembling the spectra of attention heads at 20B (Fig. 2(b), top panel, Appendix Fig. 10)—for any pre-train token budget. However, we note that the power law (PL) tail already forms after 20B pre-train, as determined by goodness-of-fit—Kolmogorov-Smirnov (KS) distance (Clauset et al., 2009) (Appendix Fig. 11, bottom panels), and the agreement with this model deteriorates with the increase in pre-train tokens and with the increase in model quality (Fig. 1). From these observations, we can conclude that the formation of a power law tail in the singular spectra is a necessary but not sufficient prerequisite for the model performance increase.

### 4.3. Spectral Evolution Along the CPT Stage—Head Heterogeneity

First, we highlight that the CPT stage does not induce significant changes in the singular spectra of model weight matrices. Apart from the similarity of matrix properties, we confirm this experimentally—namely, the model quality after CPT does not change after transplanting singular spectra from $W^{pre-train}$ into $W^{domain}$ (see Appendix Fig. 15). This allows us to investigate the effects of CPT by consid-

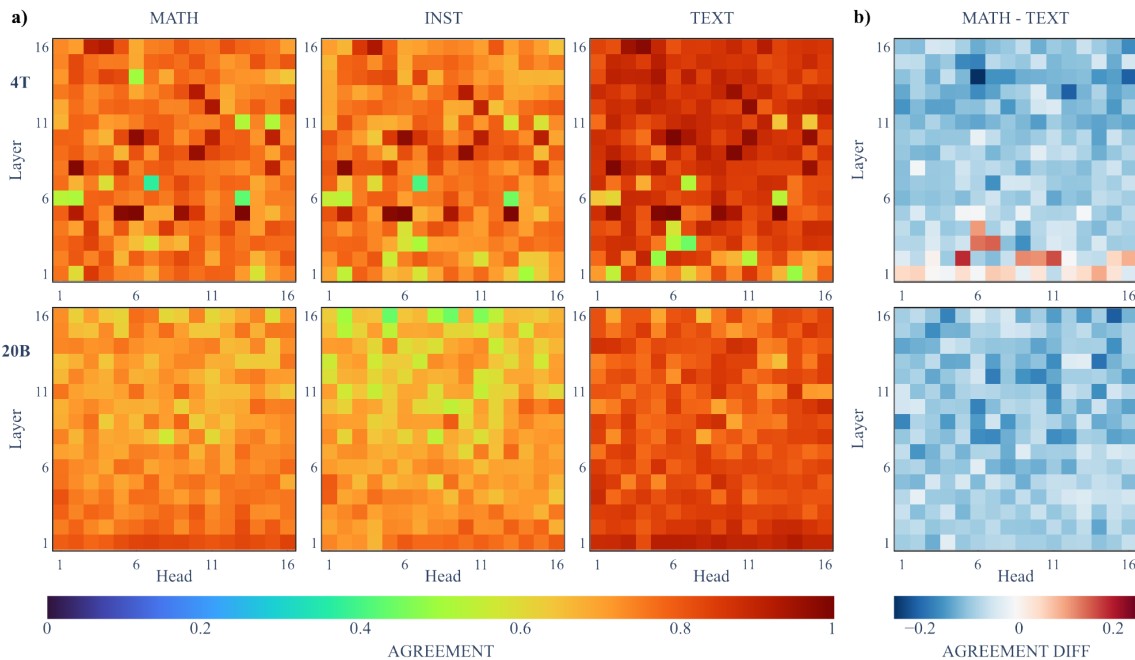

*Figure 3.* **Head heterogeneity**—effect of the CPT domain on vector agreement with the pre-trained model vs. pre-train budget: 4T tokens (top) and 20B (bottom). a) Heatmaps show mean CPT–pre-train vector agreement for individual heads of $W^Q$ across domains (math, instruct, text). Lower agreement indicates larger weight rotations from pre-train. b) Agreement difference (math minus text): blue/red indicate larger changes under math/text CPT. With larger pre-train budgets, outlier heads emerge (a) and become domain-specialized (b). Text CPT preserves the strongest agreement; math and instruct induce larger rotations.

ering only row-maximum singular vector agreement (Appendix A.3).

Vector agreement analysis can be carried out in two ways, supported by the Diffract package. First, for each singular value in the spectrum one can consider the agreement of the corresponding vector in $W^{\text{pre-train}}$ and $W^{\text{domain}}$ (Fig. 2(b)). This mode allows us to infer that the most significantly changing vectors are associated with peaks in singular value spectra. Next, one can consider agreement in an aggregated mode over all vectors in a matrix—this enables the quantification of the degree of change for each attention head or MLP layer in the model (Fig. 3(a)). We apply this analysis to study the behavior of attention heads after CPT on the domains of math, instruct, and text, and establish the **head heterogeneity** effect. Namely, for the longer 4T pre-train, attention heads change differently during CPT: some heads exhibit substantial changes during CPT across all domains, while others change in a domain-specific manner. This is shown in Fig. 3(a) for 1B model, Appendix Fig. 12(a) for 7B model, and Appendix Fig. 13(a) for 13B model. In contrast, for the shorter 20B pre-train, the changes are more uniformly distributed across heads. The onset of head heterogeneity occurs simultaneously with the increase in complexity of attention heads singular spectra; we hypothesize that both phenomena are related to the increase in CPT quality and more rapid improvement with CPT token budget.

In order to further quantify the dynamics induced by CPT we consider the Frobenius norm of the CPT delta $\Delta W = W^{\text{domain}} - W^{\text{pre-train}}$. A comparative analysis of CPT deltas across the same domains of math, instruct, and text reveals highly consistent behavior of Frobenius norms for 1B (Fig. 14(a)), 7B and 13B (Appendix Fig. 14(b)) models. For MLP layers, we observe larger changes in later layers—a trend consistent across domains. We note that such layer specialization arises with the increase in pre-train budget (Fig. 14(a)). For attention matrices, we note the high within-layer variance of CPT delta, consistent with the head heterogeneity observation above. Next, we turn to quantifying the effect of head heterogeneity on model quality after CPT.

### 4.4. Drivers of Domain Quality in CPT Deltas

In order to quantify the effect of head heterogeneity on model quality after CPT, we carry out head-wise rewind analysis and investigate CPT delta compressibility by truncating its singular spectrum.

**Head-wise rewind.** We assess the importance of individual attention heads for CPT quality as follows—we order all heads by one of the scalar importance criteria defined below and then incrementally rewind the heads in that order to the pre-train state.

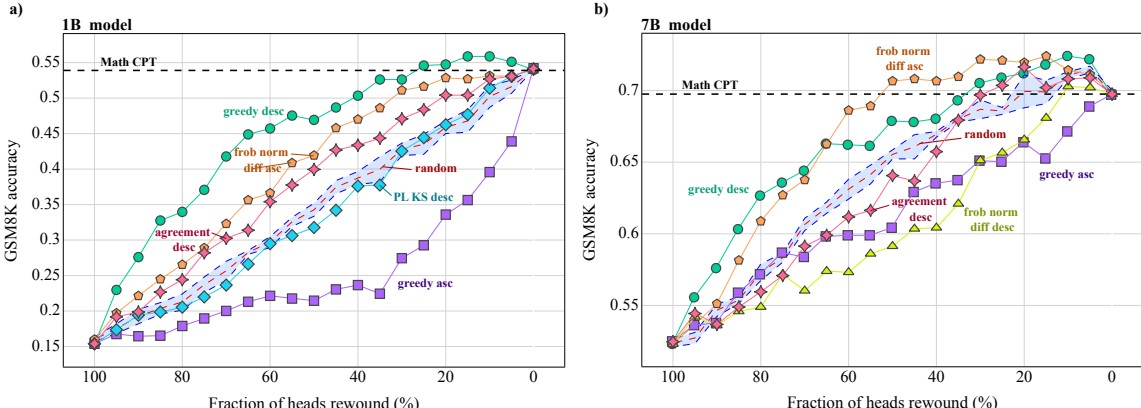

*Figure 4.* CPT delta head-wise rewind results. a) GSM8K accuracy versus fraction of heads rewound for the 1B model for different head ordering schemes: average singular vector agreement between CPT and pre-train (solid red), PL-KS distance (blue), and random (red dashed line with std band). Greedy ordering (green for descending and purple for ascending) is provided as a baseline. b) GSM8K accuracy versus fraction of heads rewound for the 7B model. The difference-in-scaled-Frobenius-norms metric, highlighted in orange (ascending), demonstrates significantly improved ordering and surpasses greedy baseline.

We evaluate the following types of ordering criteria. As a simple baseline, we rewind heads several times in different random orders. Next, we use the decrease in CPT quality upon the single-head rewind as a "greedy" ordering criterion. We treat this ranking as a "ground truth" reference measure of head importance. Additionally, we consider head orderings based on spectral properties of the corresponding weight matrices of model checkpoints and CPT deltas. For pre-train checkpoints, we use PL–KS distance and the PL exponent $\alpha$; for CPT deltas, we use the Frobenius norm and the singular vector agreement between $\boldsymbol{W}^{\text{pre-train}}$ and $\boldsymbol{W}^{\text{domain}}$.

We study CPT on math domain for 1B and 7B models after 4T pre-train, shown in Fig. 4 for 1B- and 7B-parameter models, and also instruct CPT for 7B model and math CPT for 1B model from a much smaller 20B pre-train, shown in Appendix Fig. 16. We further extend this analysis to larger-scale settings, including our 13B math and instruct CPT runs with a 10B-token CPT budget starting from the 5T pre-train checkpoint (Appendix Fig. 17), the released OLMo 2 13B CPT checkpoints trained from 5T pre-train tokens, and the released 32B OLMo 2 CPT checkpoints trained from 6T pre-train tokens (Appendix Fig. 18). Across all these settings, we observe qualitatively similar behavior. This multi-domain analysis reveals three main observations. First, rewinding any single head changes CPT quality by less than two percentage points on average, and for some heads even improves it (see Appendix Fig. 19). Second, among the simple matrix properties, performance is largely comparable to the random baseline and is outmatched by the "greedy" strategy, suggesting that head-wise spectral metrics are not very useful for assessing head importance. Finally, we demonstrate additional evidence for the emergence of head heterogeneity along the pre-train stage. Rewinding the heads

of a model with much smaller 20B pre-train budget results in a rapid decline in quality (Appendix Fig. 16(b)), consistent with the homogeneous pattern of head-wise changes during CPT (Fig. 3(a), lower panel).

Motivated by the head heterogeneity concept introduced in the previous section (Fig. 3), we propose a novel head ordering criterion that allows us (i) to achieve a quality increase—of up to **+4**% for math CPT of a 7B model upon rewinding around 15% of heads, and (ii) to rewind up to **60**% of heads without a significant quality drop. Specifically, we define the text CPT as a *reference* domain, since it is carried out on text data similar to the pre-train stage, while other CPTs are considered as *target* domains that focus on specialized knowledge. For each target domain, we order heads by the amount of change during its CPT compared to the reference CPT—such as the quantity demonstrated in Fig. 3(b). However, for 7B and 13B models, we find that a similar metric based on Frobenius norms of CPT deltas, per the equation below, performs better:

$$
\begin{aligned}
s_{l,h} = \text{scale}_{[0,1]}\Big( \|\boldsymbol{W}_{l,h}^{\text{domain}} - \boldsymbol{W}_{l,h}^{\text{pre-train}}\|_F \Big) \\
- \text{scale}_{[0,1]}\Big( \|\boldsymbol{W}_{l,h}^{\text{reference}} - \boldsymbol{W}_{l,h}^{\text{pre-train}}\|_F \Big).
\end{aligned}
\tag{1}
$$

where $l$ and $h$ index layer and head, respectively, and $\text{scale}_{[0,1]}$ denotes min–max normalization, $\text{scale}_{[0,1]}(\boldsymbol{X}) = (\boldsymbol{X} - X_{\min})/(X_{\max} - X_{\min})$.

Empirically, this novel methodology helps elucidate domain-specific changes via a comparison with a reference text domain, and outperforms not only standard spectral heuristics, but also the greedy ranking strategy (see Appendix Table 4). We hypothesize that further research of more complex models for singular spectra reflective of the rich

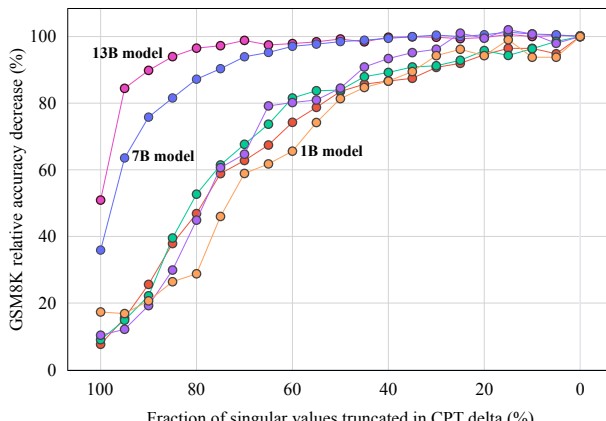

*Figure 5.* CPT delta SVD redundancy analysis. GSM8K relative accuracy as a function of the fraction of singular values retained after SVD truncation of the CPT delta. Curves show models with different scales and pre-train budgets: 13B/5T tokens (pink), 7B/4T tokens (blue), 1B/4T tokens (red), 1B/400B tokens (green), 1B/100B tokens (purple), 1B/20B tokens (orange). The 13B and 7B models retain quality upon aggressive truncation (70% and 50% of singular values removed, respectively), while 1B models degrade more rapidly, suggesting the larger model learns more redundant representations during CPT.

structure observed in Fig. 2(b), will enable the development of more powerful head ranking criteria.

**SVD truncation of CPT delta.** To assess CPT parameter redundancy, we analyze the low-rank structure of the CPT delta. For each matrix, we compute an SVD of $\Delta W$, zero out the smallest singular values, reconstruct a truncated $\Delta \widetilde{W}$, and evaluate $W^{\text{pre-train}} + \Delta \widetilde{W}$. Truncation is applied only to attention (head-wise) and MLP matrices; all other matrices are taken directly from $W^{\text{domain}}$.

For the 1B model, CPT deltas remain high-rank, and truncation tolerance does not improve monotonically with the pre-train token budget. More concretely, truncating 30% of the CPT delta leads to a relative drop of $\approx 10\%$ in GSM8K accuracy for both the 100B-token and 4T-token pre-trained models. In contrast to the token budget, model scale has a substantial effect on truncation tolerance, suggesting that larger architectures accommodate more redundant task directions. As shown in Fig. 5, for the 7B model one can remove up to 50% of singular values without a measurable drop in GSM8K accuracy, whereas for the 13B model up to **70**% can be removed; pronounced degradation, however, begins beyond 80% singular value removal for the 7B model and beyond 90% for the 13B model.

### 4.5. Domain Connectivity

Inspired by linear mode connectivity (Frankle et al., 2020) in fine-tuning of large language and vision models, we observe a similar phenomenon for continual pre-training. We study

**domain connectivity** between pairs of models initialized from the same $W^{\text{pre-train}}$ checkpoint and continually pre-trained on different domains: math, text, code, and instruct for 20B tokens, considering both 1B and 7B models, and additionally between 13B math and instruct checkpoints obtained with a 10B-token CPT budget. Specifically, we form interpolants:

$$W^{\text{interp}}(\omega) \;=\; (1-\omega)\, W^{\text{dom}_1} \;+\; \omega\, W^{\text{dom}_2} \qquad \omega \in [0, 1] \tag{2}$$

which we refer to as "*model soups*". Additionally, we evaluate models trained on different domain mixtures (see Appendix Table 2). We present results for math – instruct and math – text interpolation for 1B, 7B, and 13B models in Fig. 6; additionally, we report results for instruct – text in Appendix Fig. 20(a) and instruct – code and math – code in Appendix Fig. 21(a, b).

For the math domain (Fig. 6), for all interpolation pairs, the model soup quality lies *below the chord* connecting the endpoints (i.e., is concave) at 20B, is approximately *linear* at 400B, becomes mildly *convex* at 4T for the 1B model, and is clearly *convex* at 4T for the 7B model, as well as at 5T for the 13B model. This trend indicates that interpolation quality improves with both the pre-train token budget and model size.

We hypothesize that the transition from concave to convex behavior may be related to the development of complex spectral structure in attention heads. Additionally, across pre-train budgets and model sizes, linear interpolation underperforms CPT trained directly on dataset mixtures. This, in turn, may limit the applicability of simple task-vector-style linear combinations, but a more systematic study is left for future work.

## 5. Related Work

**Continual pre-training.** A primary challenge in CPT is mitigating catastrophic forgetting while efficiently adapting models to new domains. To combat forgetting directly, interleaving a small fraction of *replay data* from the original pre-train distribution during CPT is a simple yet powerful method to anchor the model's general representations (Wang et al., 2024; Qi et al., 2025; Hickok, 2025). The optimization process itself is crucial, as empirical findings demonstrate that learning rate re-warming and re-decaying is necessary to overcome initial instability and adapt optimization dynamics to the new data, even in the absence of a distribution shift (Gupta et al., 2023; Ibrahim et al., 2024). Furthermore, data selection strategies that choose samples based on their similarity to the target task or their novelty and diversity are highly effective for adaptation (Xie et al., 2024; Que et al., 2024).

**Model weight spectrum interventions.** Weight matri-

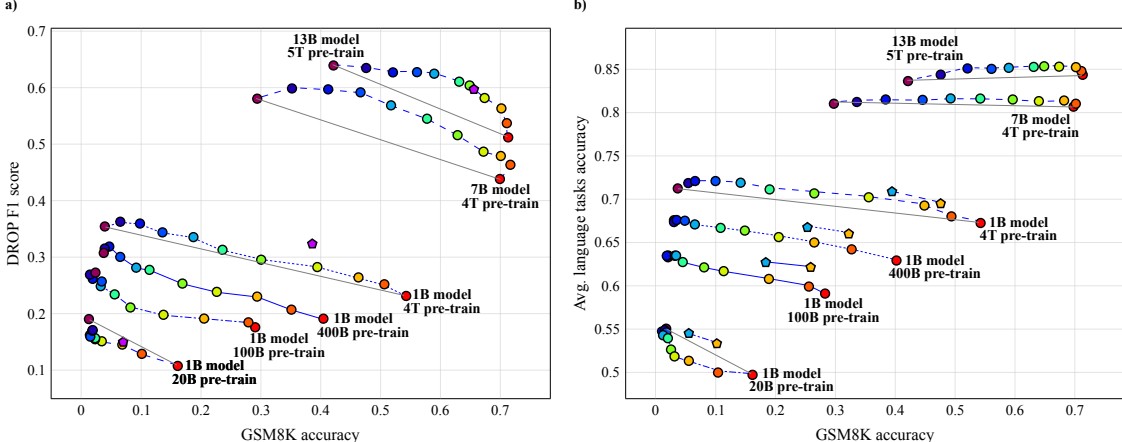

*Figure 6.* **Domain connectivity** demonstration. Quality of linear interpolation between pairs of checkpoints on different domains, step size $\omega = 0.1$. a) DROP F1 score vs. GSM8K accuracy for different pre-train token budgets and model sizes. The purple pentagon marker denotes models trained on 10B DolminoMath + 10B FLAN: from bottom to top, they correspond to the 1B model with 20B pre-train tokens, the 1B model with 4T pre-train tokens, and the 7B model with 4T pre-train tokens. b) Average language-task accuracy vs. GSM8K accuracy for different pre-train token budgets and model sizes. Yellow markers correspond to 80% Math / 20% filtered DCLM; blue markers correspond to 40% Math / 60% filtered DCLM. As the pre-train token budget and model size increase, both interpolation quality and the performance of models trained on data mixtures improve.

ces are often approximately low rank, enabling selective removal of higher-order components (small singular values) to denoise networks and enhance reasoning performance (Sharma et al., 2024). This low-rank property underpins parameter-efficient fine-tuning methods like LoRA (Hu et al., 2022). However, optimal rank is highly layer-dependent, and smaller residual singular values are not mere noise—they maintain connectivity to good loss basins and preserve performance on difficult tasks (Yin et al., 2024).

**Model averaging and task arithmetic.** Model merging is rooted in linear mode connectivity (Frankle et al., 2020), which posits that independently trained networks can lie in a shared low-error basin, enabling weight interpolation without catastrophic loss (Ainsworth et al., 2023). Model souping averages weights of models fine-tuned from a common checkpoint (Wortsman et al., 2022), and merging models from diverse training runs boosts out-of-distribution generalization (Rame et al., 2022). Task arithmetic (Ilharco et al., 2023) reframes fine-tuning as additive task vectors in nearly orthogonal directions, with TIES-Merging mitigating interference by pruning small updates and enforcing consensus signs (Yadav et al., 2023). Complementary approaches are based on the exclusion of a large proportion of delta entries from merging to confine task deltas (Yu et al., 2024; He et al., 2025), and compressing per-layer deltas via SVD with Procrustes alignment to reduce subspace overlap (Gargiulo et al., 2025).

**Spectral analysis and model performance.** RMT studies identify heavy-tailed self-regularization (HTSR) as a key feature of well-trained models, where lower power law exponents correlate with superior generalization in Transformers, serving as data-free quality predictors (Yang et al., 2023; Mahoney & Martin, 2019; Martin & Mahoney, 2021; Kothapalli et al., 2025). Training yields HTSR shaped by the dynamics of the optimizer and consistently decreasing effective rank during training across architectures, with better generalizing solutions exhibiting lower effective rank (Yunis et al., 2024; Thamm et al., 2024; Staats et al., 2024). The Marchenko–Pastur (MP) law helps distinguish bulk eigenvectors—which are largely random—from the top singular components, which encode learned signal (Thamm et al., 2024; Staats et al., 2024).

## 6. Discussion and Directions for Further Work

Our analysis allowed us to unveil several novel observations about continual pre-training (CPT) mechanisms in large language models. The properties we establish are consistent across model scale within the OLMo family, as demonstrated on OLMo 2 1B, 7B, 13B, and 32B parameter models, and diverse CPT domains, including math, instruct, code, and text.

Leveraging our Diffract framework for spectral analysis, we observe that pre-train stage induces complex spectral structures in attention-related weight matrices. While MLP layer characteristics align partially with heavy-tailed self-regularization (HTSR) theory, attention matrices display multi-peak distributions with outliers that substantially di-

verge from HTSR model. We hypothesize that this intricate spectral organization acquired during pre-train stage is essential for efficient domain adaptation during CPT. Support for this hypothesis arises from our finding that CPT largely preserves the singular value spectra and the adaptation takes place primarily via modifying the singular vectors, with the most pronounced changes occurring for vectors associated with the spectral peaks.

Our findings regarding CPT adaptation reveal a novel phenomenon in attention weight matrices which we term **head heterogeneity**. This phenomenon appears as the emergence of a small number of outlier heads in terms of CPT delta characteristics, and we identify heads whose changes are domain-specific. This feature becomes more pronounced with the increase of pre-train token budget. Based on this observation, we propose a principled criterion for ranking attention heads, identifying those that can be rewound with quality improvements of up to **4**%. Additionally, we demonstrate that CPT delta can be sparsified by up to **60**% over attention heads without significant quality loss.

Moreover, we identify another novel phenomenon—CPT **domain connectivity**, namely, the ability to average checkpoints after CPT on different domains, with interpolated model quality increasing as a function of the pre-train token budget and model size. However, we note that the mixture of CPT checkpoints performs worse compared to training on dataset mixture; these results suggest that task vector merging approaches such as DARE (Yu et al., 2024) would face significant challenges.

These findings, however, come with several limitations. While our conclusions are supported by consistent evidence across OLMo 2 1B, 7B, 13B, and 32B models, they should be interpreted as robustness within the OLMo family rather than universality across modern LLMs. Moreover, not all phenomena are studied uniformly at every scale, and our analysis is restricted to the AdamW-based OLMo 2 training recipe. Although we extend the evaluation with MATH-500 and SQuAD, benchmark coverage remains limited. We therefore view validation on other model families, broader benchmark coverage, larger-scale experiments beyond 32B, and optimizer ablations as important directions for future work.

Beyond these limitations, our results suggest several avenues for future research. First, we highlight the necessity of developing more complex models for singular value spectral of modern large language models. Based on that, a more detailed analysis of head heterogeneity and head influence on CPT quality can be carried out. Finally, we suggest further investigation into the origins of domain connectivity and its quality drivers in order to enable more efficient CPT methodology.

## Impact Statement

This work contributes to the fundamental understanding of Large Language Model training dynamics, specifically within the regime of continual pre-training. There are many potential societal consequences of our work, none of which we feel must be specifically highlighted here.

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

# A. Diffract methodology and technical details

To systematically analyze the spectral characteristics and singular vector agreement adjustments in continual pre-training, we developed Diffract, an open-source library designed for granular, architectural-component-level analysis of neural networks. The main text of the paper contains our key findings. This appendix provides a brief, practical overview of how to use our library to perform similar analyses.

## A.1. Norms

Our investigation includes the Frobenius ($\|\boldsymbol{W}\|_F$) and spectral ($\|\boldsymbol{W}\|_2$) norms:

$$\|\boldsymbol{W}\|_F = \sqrt{\sum_i \sigma_i(\boldsymbol{W})^2}, \tag{3}$$

$$\|\boldsymbol{W}\|_2 = \max_i \sigma_i(\boldsymbol{W}). \tag{4}$$

## A.2. Ranks

We track the following structure-aware ranks:

*Stable Rank:*

$$R^s(\boldsymbol{W}) = \frac{\|\boldsymbol{W}\|_F^2}{\|\boldsymbol{W}\|_2^2}. \tag{5}$$

*Effective Rank:*

$$R^e(\boldsymbol{W}) = -\sum_{i=1}^{R(\boldsymbol{W})} \frac{\sigma_i(\boldsymbol{W})}{\sum_j \sigma_j(\boldsymbol{W})} \log\left(\frac{\sigma_i(\boldsymbol{W})}{\sum_j \sigma_j(\boldsymbol{W})}\right), \tag{6}$$

where hard rank $R(\boldsymbol{W})$ is the number of nonzero singular values (or a chosen truncation).

## A.3. Singular vector agreement

Let $\boldsymbol{W}^i = \boldsymbol{U}^i \boldsymbol{\Sigma}^i \left(\boldsymbol{V}^i\right)^\top$ and $\boldsymbol{W}^j = \boldsymbol{U}^j \boldsymbol{\Sigma}^j \left(\boldsymbol{V}^j\right)^\top$. In this work we consider the cosine similarities between *left* singular vectors via the *vector agreement* matrix

$$\boldsymbol{A}^u(\boldsymbol{W}^i, \boldsymbol{W}^j) = |\left(\boldsymbol{U}^i\right)^\top \boldsymbol{U}^j|, \tag{7}$$

where the absolute value is taken element-wise. We report two levels: per-vector (diagonal and row-maximum agreements) and a global average given by the mean of diagonal agreements across the matrix.

## A.4. Random matrix theory fits

The HTSR (Heavy-Tailed Self-Regularization) theory originally emerged as a semi-empirical theory, and early seminal works (Mahoney & Martin, 2019; Martin & Mahoney, 2021) studied the empirical spectral density of weight matrices given by

$$\mu(\lambda; \boldsymbol{X}^i) = \frac{1}{n} \sum_{j=1}^n \delta\left(\lambda - \lambda_j(\boldsymbol{X}^i)\right). \tag{8}$$

Here $\boldsymbol{X}^i$ is a correlation matrix, defined as $\boldsymbol{X}^i = \frac{1}{m}\left(\boldsymbol{W}^i\right)^\top \boldsymbol{W}^i$. The eigenvalues of $\boldsymbol{X}^i$ provide insight into the distribution of information across different directions in the feature space. This is captured by the Empirical Spectral Density (ESD), where $\lambda_1(\boldsymbol{X}^i) \leq \ldots \leq \lambda_n(\boldsymbol{X}^i)$ are the eigenvalues of $\boldsymbol{X}^i$, and $\delta$ denotes the Dirac delta function. The ESD thus represents a probability measure describing how the eigenvalues are distributed. Note the relation between singular values and correlation eigenvalues: $\lambda_i(\boldsymbol{X}) = \frac{1}{m} \sigma_i(\boldsymbol{W})^2$.

At random initialization weights are modeled as entries drawn from a Gaussian Orthogonal Ensemble (GOE)

$$W_{i,j} \sim \mathcal{N}(0, \mathrm{Var}(\boldsymbol{W})). \tag{9}$$

One of the key observations in modern DNNs is the deviation of ESDs from classical RMT predictions for such matrices, such as the Marchenko–Pastur (MP) distribution

$$\rho^{\mathrm{MP}}(\lambda; \boldsymbol{X}) = \begin{cases} \dfrac{n}{2\pi m \cdot \mathrm{Var}(\boldsymbol{W})} \dfrac{\sqrt{(\lambda^{\max} - \lambda)(\lambda - \lambda^{\min})}}{\lambda}, & \lambda \in [\lambda^{\min}, \lambda^{\max}], \\ 0, & \text{otherwise}, \end{cases} \tag{10}$$

with

$$\lambda^{\max/\min}(\boldsymbol{X}) = \mathrm{Var}(\boldsymbol{W})\big(1 \pm \sqrt{n/m}\big)^2. \tag{11}$$

While random GOE matrices follow the classical MP distribution, trained neural network weight matrices deviate significantly from this behavior (Mahoney & Martin, 2019; Martin & Mahoney, 2021). During training, a non-random (signal) component typically emerges outside the MP bulk. We delineate bulk versus outliers by estimating an upper edge $\widehat{\lambda}^{\max}$ via a rescaled MP heuristic (Martin & Mahoney, 2021). To be more precise, for a particular matrix $\boldsymbol{W}$ we:

1. Randomise $\boldsymbol{W}$ by permuting its elements.

2. Compute the empirical scale $s(\boldsymbol{W})$ from the randomized matrix.

3. Find $\lambda^{\max}$ based on the MP prediction with $s(\boldsymbol{W})$.

4. Correct the scale

$$\widehat{s}^2 = s(\boldsymbol{W})^2 - \frac{1}{n} \sum_{i=1}^{n} \mathbf{1}_{\{\lambda_i > \lambda^{\max}\}} \, \lambda_i, \tag{12}$$

then find $\widehat{\lambda}^{\max}$ based on $\widehat{s}$.

Well-trained models further develop heavy tails with $\rho^{\mathrm{PL}}(\lambda; \boldsymbol{X}) = c \cdot \lambda^{-\alpha}$ with normalization constant $c$ and scaling exponent $\alpha \in (1.5, 5)$ (Clauset et al., 2009); smaller $\alpha$ indicates stronger correlations and, empirically, higher model quality (Martin & Mahoney, 2021).

We estimate $\alpha$ by Maximum Likelihood Estimation (MLE) estimator (Clauset et al., 2009); empirical evidence shows MLE performs well for $\alpha \in [1.5, 3.5]$ (Martin & Mahoney, 2021), a typical range for DNN weight matrices. Fit quality is checked with the Kolmogorov–Smirnov (KS) distance between the empirical spectrum CDF $S(\lambda)$ and the fitted PL CDF $\widehat{S}(\lambda)$

$$\mathrm{KS} = \sup_{\lambda \in \{\lambda_i(\boldsymbol{W})\}} \big| S(\lambda) - \widehat{S}(\lambda) \big|. \tag{13}$$

# B. Training and Reproducibility Details

## B.1. Data

For pre-train, we use DCLM (Li et al., 2024) sample with 100B tokens in 4 mixes: 20B DCLM sample, 100B DCLM, 200B DCLM (oversampled), and 400B DCLM (oversampled).

*Table 1.* Dataset composition for Dolmino High Quality Subset and Dolmino Math Mix.

| Source | Type | Tokens | Words | Bytes | Docs |
|---|---|---|---|---|---|
| **Mid-Training Dolmino High Quality Subset** | | | | | |
| DCLM-Baseline *FastText top 7% FineWeb > 2* | High quality web | 752B | 670B | 4.56T | 606M |
| FLAN *from Dolma 1.7 decontaminated* | Instruction data | 17.0B | 14.4B | 98.2B | 57.3M |
| **High quality total** | | **832.6B** | **739.8B** | **5.09T** | **710.8M** |
| **Mid-Training Dolmino Math Mix** | | | | | |
| TuluMath | Synthetic math | 230M | 222M | 1.03B | 220K |
| Dolmino SynthMath | Synthetic math | 28.7M | 35.1M | 163M | 725K |
| TinyGSM-MIND | Synthetic math | 6.48B | 5.68B | 25.52B | 17M |
| MathCoder2 Synthetic *Ajibwa-2023 M-A-P Matrix* | Synthetic math | 3.87B | 3.71B | 18.4B | 2.83M |
| Metamath *OWM-filtered* | Math | 84.2M | 76.6M | 741M | 383K |
| CodeSearchNet *OWM-filtered* | Code | 1.78M | 1.41M | 29.8M | 7.27K |
| GSM8K *Train split* | Math | 2.74M | 2.00M | 25.3M | 17.6K |
| **Math total** | | **10.7B** | **9.73B** | **45.9B** | **21.37M** |

For CPT we use data from FLAN decontaminated dataset, Dolmino High Quality Subset and DolminoMath Mix, as proposed in OLMo 2 (Walsh et al., 2025), this data consists of language presented in the DCLM baseline. This is filtered by FastText and the FineWeb version of the original DCLM (Li et al., 2024). For code dataset we use StarCoder (Li et al., 2023). All mixes used in CPT are presented in Table 2.

## B.2. Models and Training Configuration

This study utilizes the OLMo 2 model architecture at four different scales: 1B, 7B, 13B, and 32B (Table 3). The architecture is based on the standard Transformer decoder (Vaswani et al., 2017) and incorporates several modern enhancements:

- Removal of bias terms

- SwiGLU activation function

- Rotary positional embeddings (RoPE) with $\theta = 500,000$

- QKV clipping

- RMSNorm normalization

- Reordered layer norm (post-norm configuration)

*Table 2.* Continual pre-training mixes. * — oversampled data

| Mix Name | Dolmino Math | DCLM Filtered | FLAN filtered | StackExchange | StarCoder |
|---|---|---|---|---|---|
| **10B Mixes** | | | | | |
| 4BDM | 4B | 6B | - | - | - |
| 8BDM | 8B | 2B | - | - | - |
| MATH | 10B | - | - | - | - |
| TEXT | - | 10B | - | - | - |
| **20B Mixes** | | | | | |
| 8BDM | 8B | 12B | - | - | - |
| 16BDM | 16B | 4B | - | - | - |
| MATH | 20B* | - | - | - | - |
| TEXT | - | 20B | - | - | - |
| INST | - | - | 10B* | 10B* | - |
| 10BInst 10BDM | 10B | - | 10B* | - | - |
| CODE | - | - | - | - | 20B* |
| **50B Mixes** | | | | | |
| 20BDM | 20B* | 30B* | - | - | - |
| 40BDM | 40B* | 10B | - | - | - |
| MATH | 50B* | - | - | - | - |
| TEXT | - | 41.5B | 8.5B | - | - |

- QK normalization

- Z-loss for training stability

All models are trained in mixed precision bfloat16. A complete description of the architecture and training methodology can be found in the original OLMo 2 paper (Walsh et al., 2025).

*Table 3.* Model architecture and training hyperparameters.

| | OLMo 2 1B | OLMo 2 7B | OLMo 2 13B | OLMo 2 32B |
|---|---|---|---|---|
| **Model Architecture** | | | | |
| Hidden Dimension | 2048 | 4096 | 5120 | 5120 |
| Number of Layers | 16 | 32 | 40 | 64 |
| Number of Attention Heads | 16 | 32 | 40 | 40 (8 for KV) |
| MLP Ratio | 8 | 5.375 | 5.4 | 10.8 |
| Activation Function | SwiGLU | SwiGLU | SwiGLU | SwiGLU |
| Normalization Type | RMS Norm | RMS Norm | RMS Norm | RMS Norm |
| Positional Encoding | RoPE ($\theta = 500,000$) | RoPE ($\theta = 500,000$) | RoPE ($\theta = 500,000$) | RoPE ($\theta = 500,000$) |
| Max Sequence Length | 4096 | 4096 | 4096 | 4096 |
| Vocabulary Size | 100,278 | 100,278 | 100,278 | 100,278 |
| **Training Configuration** | | | | |
| Global Batch Size | 512 | 1024 | 2048 | 2048 |

**Pre-train stage.** For our training setup, we adhere to the parameters proposed in the original OLMo 2 paper: a learning rate of $4 \times 10^{-4}$ with a warmup phase over 0.7 billion tokens, followed by a cosine learning rate scheduler that decays to 10% of the initial rate by the end of training. The optimization is carried out using the AdamW optimizer (Loshchilov & Hutter, 2019) with the following hyperparameters: $\beta_1 = 0.9$, $\beta_2 = 0.95$, and $\epsilon = 10^{-8}$ . A weight decay of 0.1 is applied to all weights, including norms and biases, but not to embeddings.

**Continual pre-training.** The hyperparameters for continual pre-training remain consistent with those from pre-train, with the exception of the learning rate. In this stage, we start with the final learning rate from pre-train, which is $4 \times 10^{-5}$, and apply a linear annealing schedule that decreases the learning rate to zero over the course of training.

**B.3. Links to Code**

We provide full source code to ensure all our experiments are reproducible. Our release includes the code for pre-train, CPT runs and commands for the OLMEs framework evaluation. The code is publicly available at: https://github.com/Risk-AI-Research/diffract-training

We conduct a deep analysis of model weights using our open-source library, Diffract https://github.com/Risk-AI-Research/diffract

# C. Additional Experimental Results

## C.1. Additional Results for Pre-Train and CPT Spectral Analysis

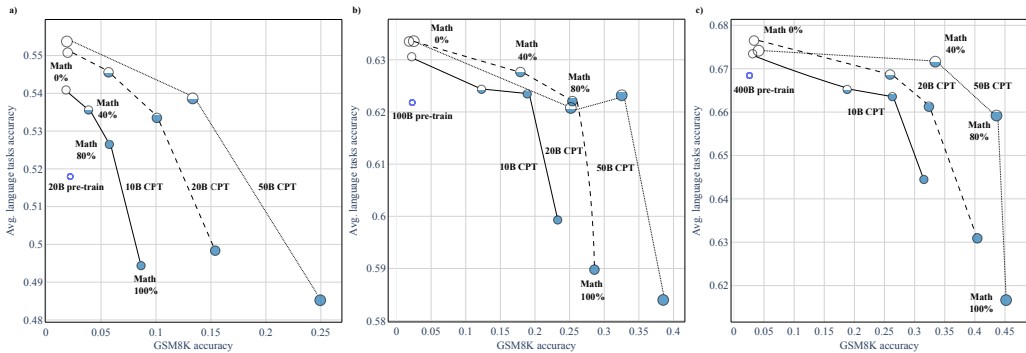

*Figure 7.* Analysis of CPT configurations varying in token budgets and math proportions. Panels (a)-(c) -show results for CPT initialized from the 20B, 100B and 400B pre-train checkpoints respectively. Performance is evaluated as a function of the mathematical data proportion in the CPT mix.

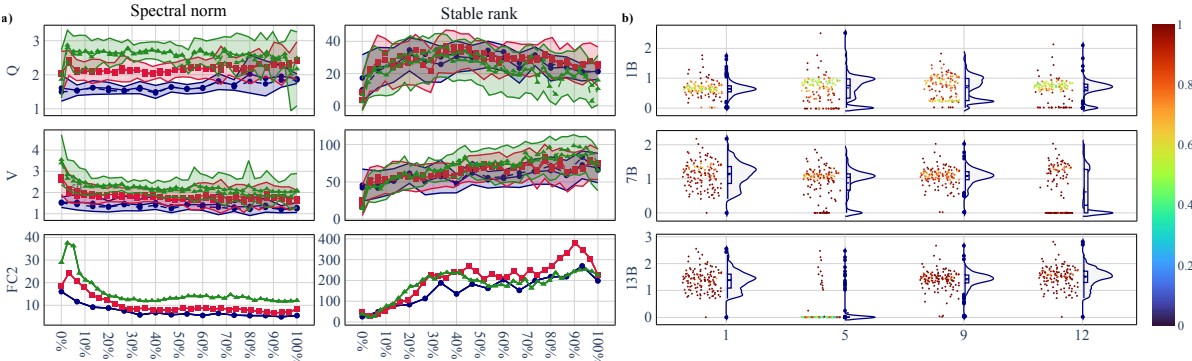

*Figure 8.* Comparison of spectral properties of 1B, 7B, and 13B parameter models. a) Layer-wise spectral norm and stable rank comparison for $W^Q$, $W^V$, and $W^{FC_2}$ matrices. The x-axis represents the relative layer depth, normalized as a percentage of total layers to account for the differing total counts (16, 32, and 40 layers for the 1B, 7B, and 13B models, respectively). Notably, the trends among model scales are qualitatively similar. b) Singular value spectra for $W^Q$ (layer 15, heads 1, 5, 9, 12) for 1B and 7B model sizes for 4T tokens, and 13B model size for 5T tokens. The spectral structures for both model sizes exhibit significant complexity and deviate from standard theoretical distributions such as MP or PL.

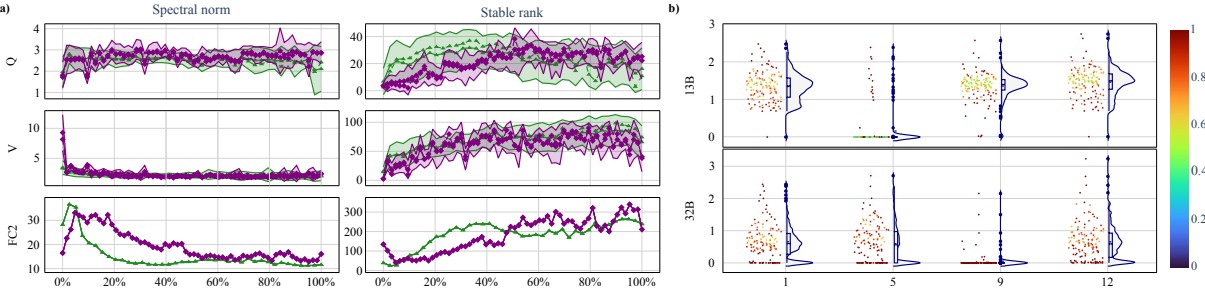

*Figure 9.* Comparison of spectral properties of 13B and 32B parameter models of OLMo 2 original CPT checkpoint. a) Layer-wise spectral norm and stable rank comparison for $W^Q$, $W^V$, and $W^{FC_2}$ matrices. The x-axis represents the relative layer depth, normalized as a percentage of total layers to account for the differing total counts (40 and 64 layers for the 13B and 32B models, respectively). Notably, the trends for both model scales are qualitatively similar. b) Singular value spectra for $W^Q$ (layer 15, heads 1, 5, 9, 12). The spectral structures for both model sizes exhibit significant complexity and deviate from standard theoretical distributions such as MP or PL.

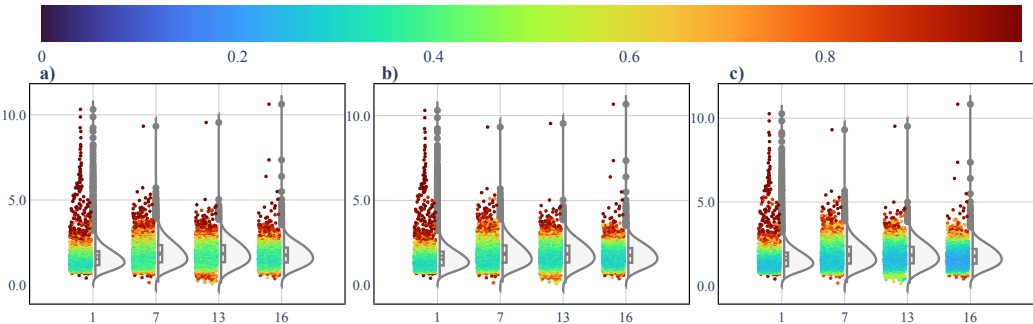

*Figure 10.* The effect of CPT data domain on vector agreement with pre-train. Violin plots (with jittered points colored by the left singular vector agreement between CPT and pre-train) display the $W^{\text{GATE}}$ per layer for CPT a) on text, b) a math-text mix, c) or pure math (all pre-trained on 4T tokens). The results demonstrate a clear ordering: text CPT preserves the strongest vector agreement, while math domain induces the largest rotation.

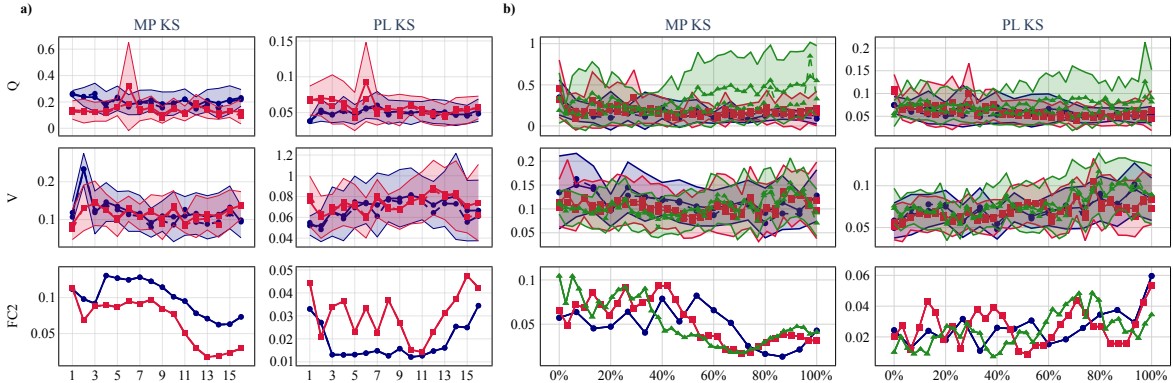

*Figure 11.* Goodness-of-fit—layer-wise Kolmogorov–Smirnov distance between the weight matrix ESD and Marchenko–Pastur model (left) and power law (right) model for $W^{\text{Q}}$, $W^{\text{V}}$, and $W^{\text{FC}_2}$ matrices. The x-axis indexes layers; solid lines show the mean across attention heads, dashed lines the median, and the shaded band denotes mean ± std. a) Blue denotes 20B-token pre-train and red denotes 400B-token pre-train of 1B model. b) Blue denotes 4T-token pre-train of 1B model, red denotes 4T-token pre-train of 7B model and green denotes 5T-token pre-train of 13B

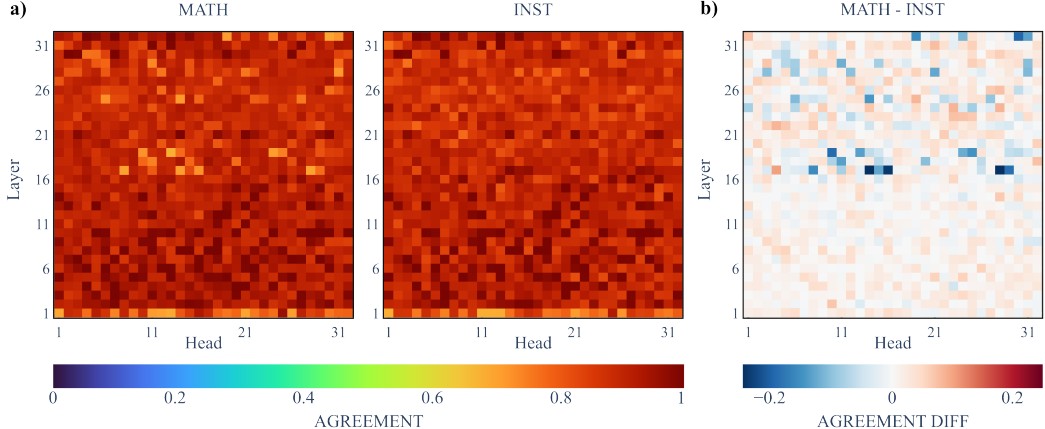

*Figure 12.* Effect of CPT data domain on vector agreement. a) Heatmaps show the average vector agreement between CPT and pre-train for individual heads of $W^{\text{Q}}$ across different CPT domains (left to right: math, instruct) on 7B model. Lower vector agreement values correspond to larger changes of the corresponding weight matrices relative to the pre-trained model. b) Difference in vector agreement between the math and instruct domains.

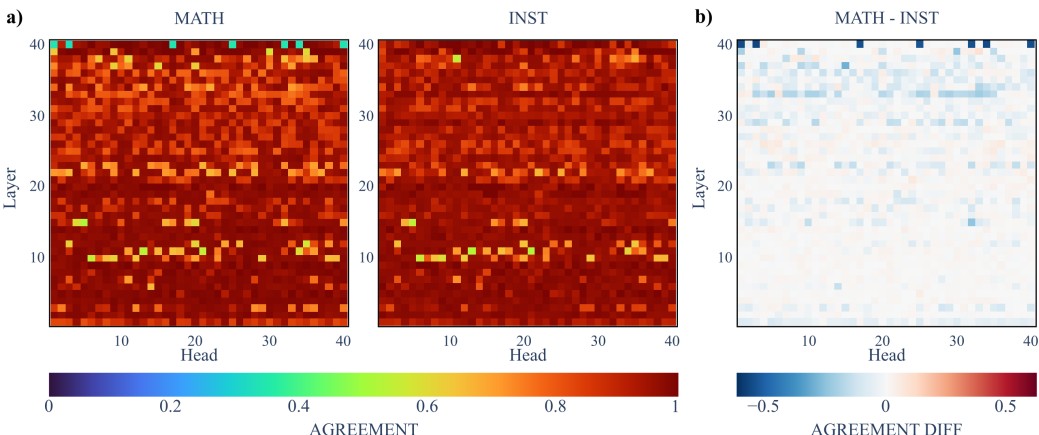

*Figure 13.* Effect of CPT data domain on vector agreement. a) Heatmaps show the average vector agreement between CPT and pre-train for individual heads of $\boldsymbol{W}^{\mathrm{Q}}$ across different CPT domains (left to right: math, instruct) on 13B model. Lower vector agreement values correspond to larger changes of the corresponding weight matrices relative to the pre-trained model. b) Difference in vector agreement between the math and instruct domains.

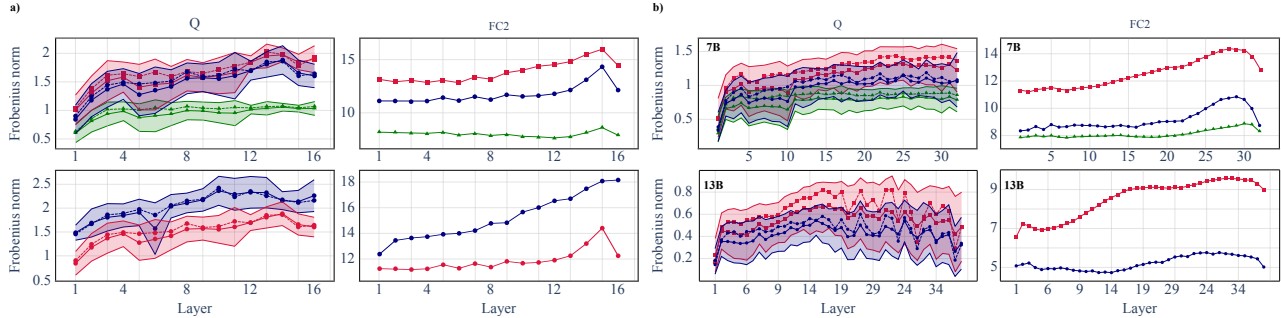

*Figure 14.* Layer-wise Frobenius norm of CPT deltas for $\boldsymbol{W}^{\mathrm{Q}}$ and $\boldsymbol{W}^{\mathrm{FC_2}}$ across (a) CPT domains and pre-train budgets for 1B model and (b) across CPT domains for 7B and 13B models. Statistics are aggregated per head (mean: solid line, median: dashed line; shaded region: mean ± std). a) For 4T-token pre-train (top), text-domain deltas (green) are smallest vs math (blue) or instruct (red) domains; in the math domain (bottom), deltas from 100B-token models (blue) exhibit a simpler layer-wise shape compared to 4T-token models (red). b) Effect of CPT domain for 4T tokens pre-train for 7B and 13B models: deltas for the text domain (green) are smallest in magnitude, indicating less incremental change compared to math (blue) or instruct (red) domains.

## C.2. Singular Value Transplantation

Given two checkpoints with per-layer weights $\boldsymbol{W}^{\mathrm{ckpt1}} = \boldsymbol{U}_1 \, \boldsymbol{\Sigma}_1 \, \boldsymbol{V}_1^{\top}$ and $\boldsymbol{W}^{\mathrm{ckpt2}} = \boldsymbol{U}_2 \, \boldsymbol{\Sigma}_2 \, \boldsymbol{V}_2^{\top}$ (SVD), the *transplanted* weight is

$$\widetilde{\boldsymbol{W}} = \boldsymbol{U}_2 \, \boldsymbol{\Sigma}_1 \, \boldsymbol{V}_2^{\top}. \tag{14}$$

Unless otherwise stated, transplantation is applied head-wise to attention $\left(\boldsymbol{W}^{\mathrm{Q}}, \boldsymbol{W}^{\mathrm{K}}, \boldsymbol{W}^{\mathrm{V}}, \boldsymbol{W}^{\mathrm{O}}\right)$ and MLP $\left(\boldsymbol{W}^{\mathrm{FC_1}}, \boldsymbol{W}^{\mathrm{Gate}}, \boldsymbol{W}^{\mathrm{FC_2}}\right)$ weight matrices only; all other parameters remain from the target checkpoint. We perform SVD transplantation from CPT to pre-train, see Figure 15.

### C.2.1. SINGULAR VALUE PERMUTATION ABLATIONS

To test whether singular values are interchangeable across spectral regions, we perform a controlled ablation on a fixed checkpoint. Given $\boldsymbol{W} = \boldsymbol{U}\boldsymbol{\Sigma}\boldsymbol{V}^{\top}$, we construct $\boldsymbol{W}_p = \boldsymbol{U} \, \boldsymbol{\Sigma}_p \, \boldsymbol{V}^{\top}$, where $\boldsymbol{\Sigma}_p$ is obtained by randomly permuting a selected subset of diagonal entries of $\boldsymbol{\Sigma}$ while keeping the remaining entries unchanged. We note that upon full spectrum shuffle model quality drops almost to zero and shuffling only bulk does not induce significant quality loss.

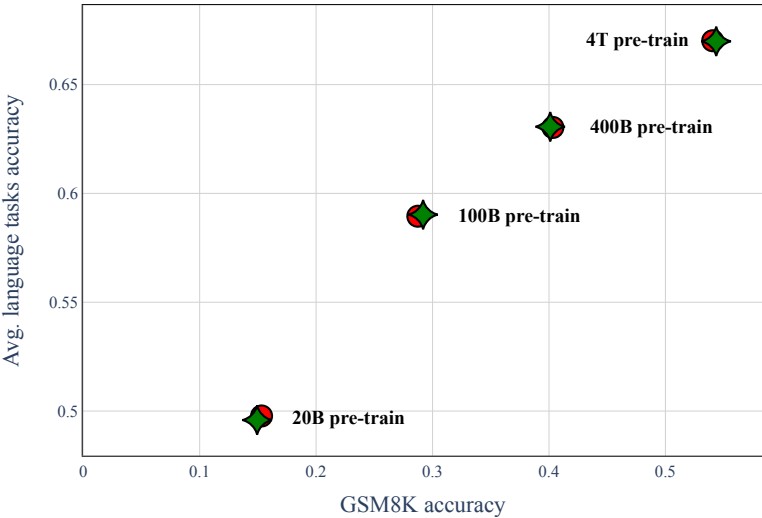

*Figure 15.* Singular value transplantation results. Red circles denote CPT on DolminoMath, while star–diamond markers denote CPT on DolminoMath with singular values transplanted from the pre-train model. The results indicate that singular value transplantation does not affect GSM8K accuracy.

### C.3. Methodology and Additional Results for Head-Wise Rewind

Let $H$ be the number of heads and $d_{\text{head}}$ the head dimension, so $d_{\text{model}} = H\, d_{\text{head}}$. We split projections along the head-concatenated axis and operate per head:

$$\boldsymbol{W}^Q = [\,\boldsymbol{W}^Q_{(0)} \mid \ldots \mid \boldsymbol{W}^Q_{(H-1)}\,], \; \boldsymbol{W}^K = [\,\boldsymbol{W}^K_{(0)} \mid \ldots \mid \boldsymbol{W}^K_{(H-1)}\,], \; \boldsymbol{W}^V = [\,\boldsymbol{W}^V_{(0)} \mid \ldots \mid \boldsymbol{W}^V_{(H-1)}\,] \tag{15}$$

where $\boldsymbol{W}^{Q/K/V}_{(i)} \in \mathbb{R}^{d_{\text{model}} \times d_{\text{head}}}$ are column blocks. For the output projection $\boldsymbol{W}^O \in \mathbb{R}^{(H\, d_{\text{head}}) \times d_{\text{model}}}$, we split by rows into $H$ blocks $\boldsymbol{W}^O_{(i)} \in \mathbb{R}^{d_{\text{head}} \times d_{\text{model}}}$ and reassemble by row concatenation. Single-head rewind at index $i$ sets

$$\{\boldsymbol{W}^Q_{(i)}, \boldsymbol{W}^K_{(i)}, \boldsymbol{W}^V_{(i)}, \boldsymbol{W}^O_{(i)}\}^{\text{math}} \leftarrow \{\boldsymbol{W}^Q_{(i)}, \boldsymbol{W}^K_{(i)}, \boldsymbol{W}^V_{(i)}, \boldsymbol{W}^O_{(i)}\}^{\text{pre-train}} \tag{16}$$

and replaces the corresponding QK-norm segments along the head axis with pre-train values.

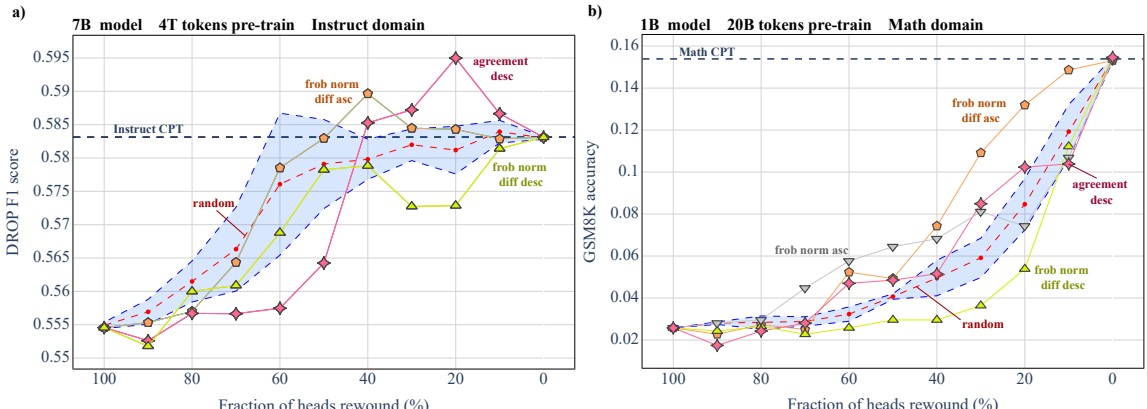

*Figure 16.* CPT delta head-wise rewind results. a) DROP F1 score versus the fraction of heads rewound for a 7B-parameter model with a 4T-token pre-train budget and a 20B-token instruct domain. Rewinding heads according to any heuristic has only a minor effect on CPT quality in the instruct domain, while random rewind exhibits high variance across different random seeds. b) GSM8K accuracy versus the fraction of heads rewound for a 1B-parameter model with a 20B-token pre-train budget and a 20B-token math domain. Here, head rewinding under all heuristics leads to a rapid and substantial degradation in performance, indicating lack of head heterogeneity at small pre-train token budgets. In both panels, our proposed difference-in-scaled-Frobenius-norms metric (highlighted in orange) yields the most effective head ranking for preserving task performance.

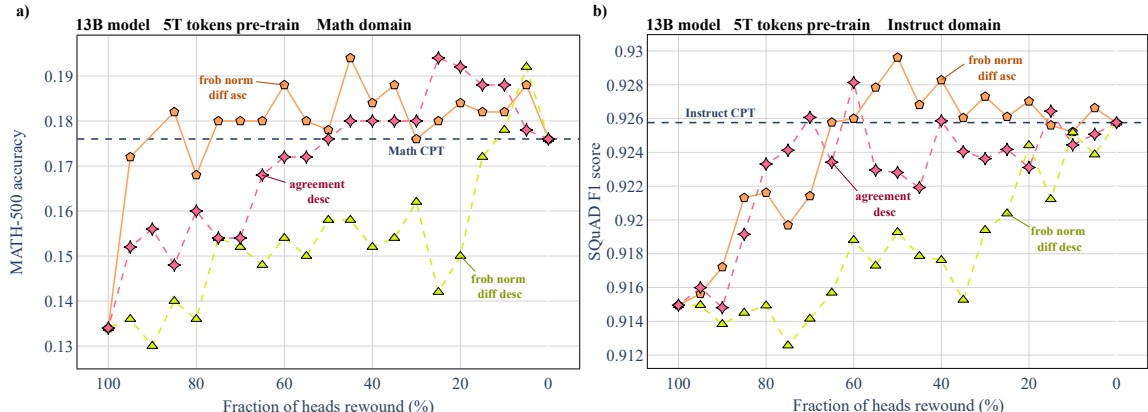

*Figure 17.* CPT delta head-wise rewind results for 13B model with 5T token pre-train for 10B token CPT. a) MATH-500 accuracy versus the fraction of heads rewound. Rewinding heads gains 2.2%. b) SQuAD F1 score versus the fraction of heads rewound. Head rewinding according to any heuristic has only minor effect on CPT quality. Moreover over 60% can be rewound without any quality loss.

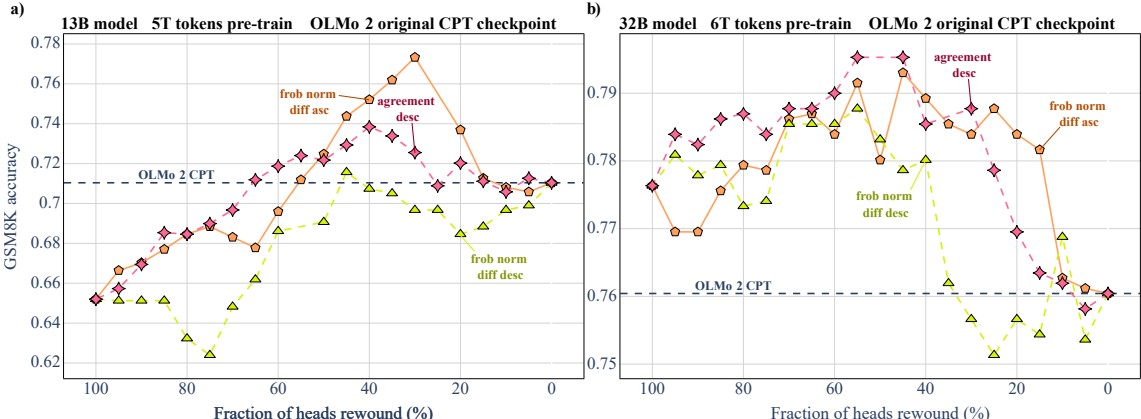

*Figure 18.* CPT delta head-wise rewind results for 13B and 32B OLMo 2 original checkpoints. a) GSM8K accuracy versus the fraction of heads rewound for 13B original OLMo 2 CPT checkpoint. Rewinding heads gains 6.3%. b) GSM8K accuracy versus the fraction of heads rewound for 32B original OLMo 2 CPT checkpoint. Head rewind gains improvements up to 3.6%. Moreover, complete head rewind is achieved with simultaneous quality improvement.

**AUC computation.** Let $\{k_t\}_{t=1}^{T}$ be the discrete percentages of heads rewound (in $[0, 100]$), and let $\mathcal{C}_\downarrow(k_t)$ and $\mathcal{C}_\uparrow(k_t)$ denote the metric at $k_t$ for descending and ascending greedy orders, respectively. We first compute the discrete AUC of each curve using the trapezoidal rule:

$$\text{AUC}_\downarrow = \sum_{t=1}^{T-1} \frac{\mathcal{C}_\downarrow(k_t) + \mathcal{C}_\downarrow(k_{t+1})}{2} \left(k_{t+1} - k_t\right), \qquad \text{AUC}_\uparrow = \sum_{t=1}^{T-1} \frac{\mathcal{C}_\uparrow(k_t) + \mathcal{C}_\uparrow(k_{t+1})}{2} \left(k_{t+1} - k_t\right) \tag{17}$$

Our reported score is the absolute difference

$$\text{AUC-diff} = \left| \text{AUC}_\downarrow - \text{AUC}_\uparrow \right| \tag{18}$$

When the grid is uniform, $k_{t+1} - k_t = \Delta$, this reduces to a constant $\Delta$ times the sum of trapezoid averages. Higher values indicate a greater separation between descending and ascending orders, while values near zero indicate random-like behavior.

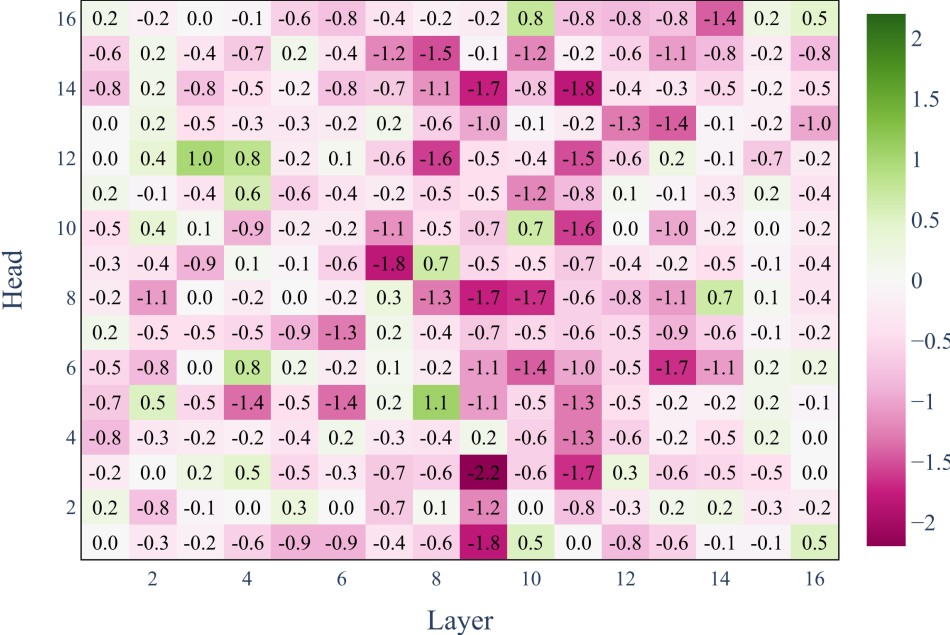

*Figure 19.* Head-wise rewind heatmap (per-layer, per-head impact) for 20B Math CPT based on 4T pre-train. Cells contain GSM8K accuracy change (%).

*Table 4.* Absolute area between descending and ascending head-rewind curves for math domain (AUC; higher is better). Smaller values indicate behavior closer to random, where ordering does not matter.

| Heuristic | AUC-diff 1B | AUC-diff 7B |
|---|---|---|
| greedy | 19.1 | 5 |
| difference in average singular vector agreement between math and text | 11.7 | 5.1 |
| difference in scaled Frobenius norms | 11 | 6 |
| agreement average | 5.2 | 4.5 |
| delta Frobenius norm | 5.2 | 4.9 |
| PL KS (pre-train) | 0.9 | 1.1 |
| random | 0.0 | 0.0 |

## C.4. CPT Delta SVD Truncation Methodology and Additional Results

Given $\boldsymbol{W} = \boldsymbol{U}\,\boldsymbol{\Sigma}\,\boldsymbol{V}^\top$ with $r = \min(m, n)$, setting $k = \lfloor (n/100)\, r \rfloor$ yields the top-$k$ truncation

$$\widetilde{\boldsymbol{W}}_{(n\%)} = \boldsymbol{U}_{[:,1:k]}\,\boldsymbol{\Sigma}_{1:k,1:k}\,(\boldsymbol{V}^\top)_{[1:k,:]}, \tag{19}$$

where we keep the top-$k$ singular directions and discard the rest. We report the kept fraction $n\%$.

## C.5. Results for Code Domain CPT

We conducted preliminary CPT experiments on code to assess domain generality, training on the StarCoder corpus and evaluating on HumanEval. The 1B model achieved 0.04 pass@1 before code CPT and only 0.09 pass@1 after code CPT—an absolute improvement of just 5 percentage points. This limited improvement aligns with OLMo-2's behavior, where competitive HumanEval performance emerges only after supervised fine-tuning with chat templates, not from base or continual pre-training.

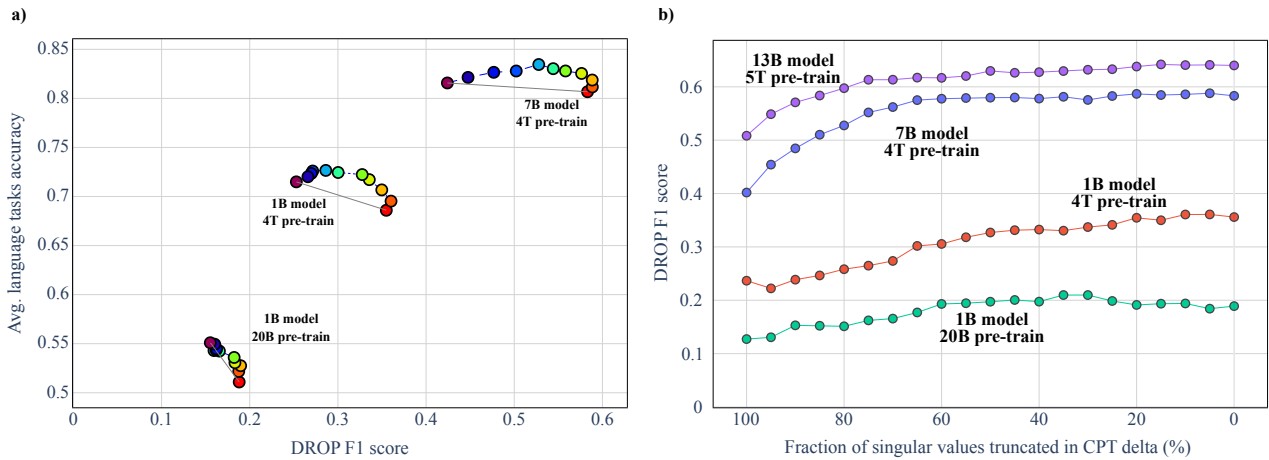

*Figure 20.* Linear interpolation and CPT delta redundancy analysis on instruct domain. a) Quality of linear interpolation, step size $\omega = 0.1$. Average language-task accuracy vs. DROP F1 score. As the pre-train token budget and model size increase, interpolation quality improves. b) CPT delta spectral redundancy analysis. DROP F1 score vs. fraction of singular values kept under CPT delta SVD truncation. Curves show models with different scales and pre-train budgets: 13B/5T tokens (violet), 7B/4T tokens (blue), 1B/4T tokens (red), 1B/20B tokens (green). A substantially larger fraction of the CPT delta can be truncated in the 13B and 7B models than in the 1B model.

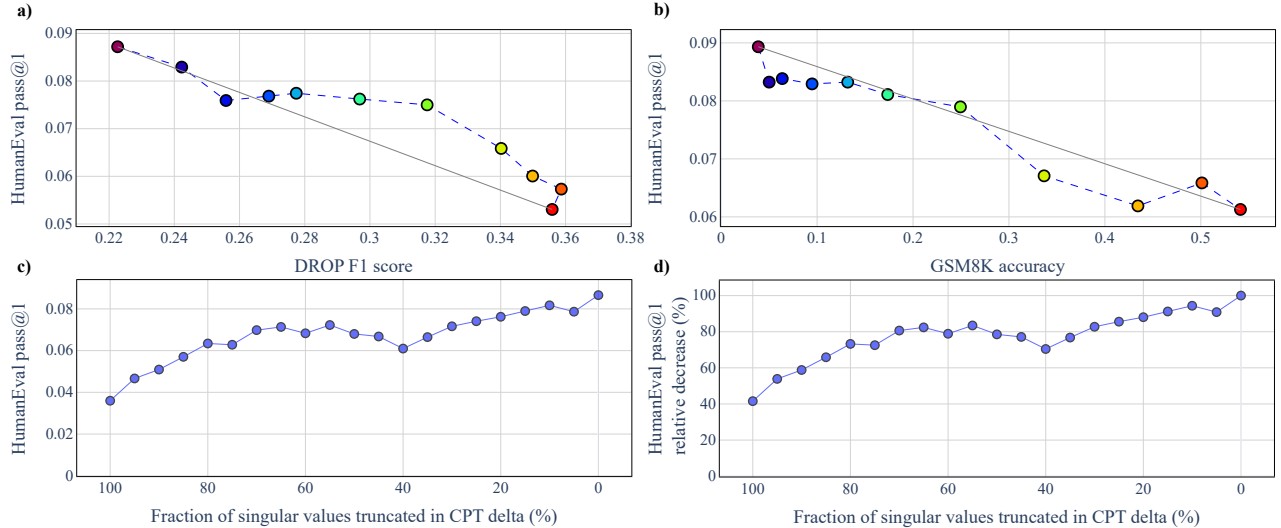

*Figure 21.* a) Quality of linear interpolation between 1B/4T tokens Instruct CPT and 1B/4T tokens Code CPT, step size $\omega = 0.1$. b) Quality of linear interpolation between 1B/4T tokens Math CPT and 1B/4T tokens Code CPT, step size $\omega = 0.1$. c) Truncation of 1B/4T tokens Code CPT delta with HumanEval pass@1 accuracy reported. d) Truncation of 1B/4T tokens Code CPT delta with relative HumanEval pass@1 accuracy drop reported.

# D. Statistical Robustness

This section specifies the uncertainty quantification protocol used in our research. The goal is to rule out artifacts caused by nondeterminism in head selection and evaluation.

**Evaluation randomness.** For metrics computation, we estimate performance variability using an 80% bootstrap procedure with 5 seeds, applied to models after 20B-token Stage 2 continual pre-training on the DolminoMath dataset. Results for both the 1B and 7B models are summarized in table 5. We observe that the variance in metrics computation is small to affect the conclusions drawn in the paper.

*Table 5.* Evaluation variability on fixed checkpoints. Entries report the mean and std from 5 seeds.

| Model | GSM8K (acc) | DROP (F1) |
|---|---|---|
| **7B** | $0.698 \pm 0.005$ | $0.437 \pm 0.015$ |
| **1B** | $0.534 \pm 0.009$ | $0.232 \pm 0.006$ |

**Head-selection variability.** Some head-ordering techniques used to choose which heads to rewind can be nondeterministic.

To quantify the sensitivity of rewinding curves to this source of randomness *independently* of training, we repeat the head-selection procedure up to 5 times with different head-selection RNG seeds on the same fixed checkpoint. For each run we recompute the full rewinding curve under identical evaluation settings, and report mean and std on the graph on Figures 4, 16.

# E. Key Diffract Visualization Tools

The pipeline consists of five sequential stages, each designed to probe different aspects of the network's weight matrices. We assume the user is comparing two model checkpoints, $W^{\text{ckpt1}}$ and $W^{\text{ckpt2}}$, and their delta, $(W^{\text{ckpt1}} - W^{\text{ckpt2}})$.

## Box plots: component-wise spectral metrics distribution

The analysis employs comparative box plots generated for all weight matrices, which are organized by matrix family—specifically, the Attention projections ($W^{\text{Q}}$, $W^{\text{K}}$, $W^{\text{V}}$, $W^{\text{O}}$) and the MLP layers. To ensure a granular analysis, each attention matrix is first decomposed into its heads before any metric computation.

The analysis itself is structured by distinct metric categories for precise comparison: one group focuses on norms, including the Frobenius and Spectral norm, while another group concentrates on rank measures, namely the Stable rank and Effective rank. For the MLP matrices exclusively, this set of metrics is augmented with the Kolmogorov-Smirnov (KS) statistic, which quantifies the fit to both a power law (PL) and a Marchenko–Pastur (MP) distribution.

The visual encoding of the box plots is designed to convey multiple data dimensions simultaneously. The fill color of each box plot signifies the specific matrix parameter. The outline color denotes the checkpoint, where blue represents ckpt1, red represents ckpt2, and green represents the delta. Furthermore, a jittered scatter plot is placed near each box; the individual data points are colored on a continuous spectral scale from blue to red. This color mapping corresponds directly to the layer index, with blue indicating layers near the model input and red indicating layers near the output. This integrated approach allows for the immediate assessment of distributional properties—including medians, quartiles, and outliers—across the two checkpoints, while preserving the crucial ability to discern depth-dependent patterns within the metric distributions.

## Cluster bar charts: singular values distribution

The purpose of this stage is to visualize the entire distribution of singular values. The visualization employs a cluster bar chart where the x-axis is exponential binning of the singular values, while the y-axis represents the count of values falling into each bin.

To articulate the layer-by-layer evolution of the spectrum, the color of each bar corresponds to its layer index. For attention matrices, the singular values are first averaged across all heads within a given layer before the binning process.

## Sparklines: spectral metric trends

This stage aims to compactly visualize the trend of each metric across the network's depth, enabling a quick, at-a-glance assessment of layer-wise patterns. The procedure involves plotting the metric value as a function of layer index for each matrix family. For attention matrices, the values are aggregated across their heads: a solid line represents the mean, a dashed line represents the median, and a shaded area delineates the band of mean ± one standard deviation. These resulting sparklines provide a temporal view of metric evolution through the network's layers. This visualization allows for the immediate identification of critical patterns, where a sharp change in a metric at a specific layer or a consistent divergence between the mean and median can signal important architectural transitions or anomalies.

## Heatmaps: spectral metric heterogeneity

The purpose of this stage is to expose fine-grained, head-specific and layer-specific variations in metrics, providing a direct visual comparison of the differences between the two checkpoints. The procedure involves creating a two-dimensional grid for each combination of attention matrix type ($W^{\text{Q}}$, $W^{\text{K}}$, $W^{\text{V}}$, $W^{\text{O}}$) and metric, with the axes representing the layer index and head index. For each such combination, a quartet of heatmaps is generated: the first visualizes the raw metric values for $W^{\text{ckpt1}}$ on a blue color scale; the second visualizes the raw values for $W^{\text{ckpt2}}$ on a red scale; the third shows the delta and the fourth displays a metric difference. This visualization allows for pinpointing the exact heads and layers that contribute most significantly to the overall divergence between the two models, moving from a high-level summary to a precise diagnostic tool.

**Violin plots: singular value distribution and singular vector agreement**

The final stage of our analysis is designed to provide the most detailed view of the singular value distribution for each individual head and layer, while simultaneously incorporating information about singular vector agreement. This is achieved by generating a matrix of plots, where each cell corresponds to a specific (layer, head) pair. Within a given cell, a violin plot is used to depict the density of the singular values for that specific matrix. Overlaid onto this violin plot is a jittered scatter plot, where the color of each point is determined by a vector agreement metric — overlap between the corresponding singular vectors of $W^{\text{ckpt1}}$ and $W^{\text{ckpt2}}$. This powerful composite visualization synergistically combines the shape of the singular value spectrum, conveyed by the violin, with the stability of the underlying directional components, indicated by the colored jitter, across the entire model. It thereby enables the precise identification of nuanced scenarios, such as attention heads that exhibit a similar spectral distribution but possess different singular vectors directions, or vice versa, offering insight into the micro-dynamics of network evolution.

