# OpenReview forum: "Diffract: Spectral View of LLM Domain Adaptation"
_ICML.cc/2026/Conference — ICML 2026 spotlight_

### Official Review · Reviewer_YNp2 · 2026-03-06

**Soundness:** 3
**Presentation:** 3
**Significance:** 3
**Originality:** 3
**Overall Recommendation:** 5
**Confidence:** 4

**Summary:**

This paper studies continual pre-training (CPT) through the lens of singular value decomposition. Experiments were performed on OLMo 2 models with 1 and 7B parameters to study the trajectory of the singular vectors and tie these to changes in model performance across code and language benchmarks. The results showed that attend heads exhibited high levels of heterogeneity and demonstrated that up to 60% of them could be removed without adversely affecting model performance. They also showed that rewinding heads with low importance (they introduce an importance measure for heads) actually improves benchmark performance, although marginally. The authors also release their toolkit for performing spectral analysis, which they name Diffract.

**Compliance With Llm Reviewing Policy:**

Affirmed.

**Final Justification:**

The author's second rebuttal mostly resolved my concerns about scaling beyond 7B parameter models by including the 32B model from the Olmo 2 family. As such, I have increased my review from Weak Accept to Accept and also increased my confidence score after carefully reading the paper again and the authors' responses.

**Key Questions For Authors:**

1. Did you evaluate this approach on other model families and larger models, and if so, were the results aligned with the conclusions presented in the paper?
2. Have you tested whether the head heterogeneity persists when using different optimizers?

**Limitations:**

No, the manuscript lacks a dedicated Limitations section and does not address the obvious limitations in generalizability, as discussed in the Strengths/Weaknesses above.

**Strengths And Weaknesses:**

**Strengths**:
1. While the SVD-based method is fairly simple, the authors presented a novel use case that reveals interesting insights into CPT, which could potentially impact the ways that foundation model developers perform domain adaptation.
2. Similar to the item above, the results have the potential to increase the efficiency of CPT. Given the energy-heavy nature of CPT, this could be impactful and beneficial to the community.
3. The evaluation and analysis on OLMo expressed a high level of depth/rigour and the visualizations aid significantly in interpretation of the results.
4. The writing style, formatting, and organization of the paper are clear and easy to read.
5. Releasing both training code and the Diffract library is appreciated.

**Weaknesses**:
1. The generalizability claims are not well-founded. Despite a rigorous analysis on OLMo 1 and 7B, the authors do not evaluate their techniques on other models. Without doing so, it is not clear how generalizable these results are to other LLMs.
2. Related to the last point, the maximum model size that was evaluated in 7B. It is not clear how these results scale to larger models, which are more prevalent in practice.
3. The manuscript is lacking a Limitations section. At the very least, the authors need to more strongly emphasize the limitations in generalizability.
4. While the writing is clear, it is very obviously heavily edited by generative AI. This is not a major weakness in my opinion, but I did find it distracting, with the writing feeling quite robotic.
5. The evaluation benchmarks are fairly limited. Math performanes is assessed only with GSM8K, instruction-following only with DROP, and code with just HumanEval. A single benchmark for each domain reduces the robustness of the claims made in the paper.
6. The relationship between head heterogeneity and optimization algorithm is unclear. It is possible that these two are coupled. A stronger ablation study on hyperparameters would help with interpretation of the significance (and mechanism behind) this observation.

---

> ### Author Rebuttal · Authors · 2026-03-31
>
> We thank the reviewer for their insightful review and overall positive assessment. We address the questions below.
>
> **Q.1; W.1**
>
> In our work we have considered only OLMo family models with up to 7B parameters for the following reasons: for our investigation of CPT runs on different domains we require the LLM checkpoint before the CPT stage; otherwise, we would need to re-run entire pipeline up to CPT, which imposes prohibitive resource requirements. This rules out popular models such as Llama [1] and Qwen [2] families. Among the models with fully open training pipeline, such as Pythia [3], SmolLM [4], Amber [5], MAP Neo [6], we have chosen to focus on OLMo, as it demonstrates very high quality, almost on par with Llama and Qwen.
>
> As for the model size, we note that 7B or similar-sized models are present in almost all modern model lineups, so this model size is relevant for practical applications. Given the limited computational resources we have chosen to carry out more experiments on this model size compared to scaling up. Additionally, our findings mostly hold for both 1B and 7B models, so we do not expect them to become invalid upon model scaling.
>
> We will include the discussion above into the "Limitations" section of the paper after the rebuttal stage. We note that extending our analysis to other model families and model scales is a natural direction for future work, and we have designed the open-source _Diffract_ library specifically to facilitate such investigations.
>
> **Q.2; W.6**
>
> We focused our study on the OLMo 2 architecture, which uses the AdamW optimizer. Based on a comprehensive survey [7], we conclude that this is de facto standard for pre‑train and CPT across modern LLMs, such as Llama 3 [1], SmolLM2 [4], Phi‑4 [8], all of which use AdamW. Thus, our choice reflects current industrial and academic practice, ensuring that our findings are directly applicable to the majority of modern LLMs.
>
> We agree that systematically investigating the effect of different optimizers (e.g., Muon [9]) on spectral evolution and head specialization is an interesting area of research. We will explicitly highlight this as a key direction in a new "Limitations" section of the paper.
>
> **W.3**
>
> We fully agree that new "Limitations" section will strengthen our work. We will add this section based on the suggestions during the rebuttal stage.
>
> **W.4**
>
> We apologize for creating an impression of AI-generated text; however we would like to reiterate that we deliberately have not used any AI tools, and that the monotonic effect is most likely due to stylistic preferences of the authors.
>
> **W.5**
>
> We have chosen to focus on GSM8k, DROP, and HumanEval benchmarks due to their widespread use for evaluating LLMs after pre-train and CPT stages. Per your suggestion we have carried out evaluations on additional benchmarks: MATH-500 [10] for math CPT and SQuAD [11] for instruct CPT. We observe that all of the key properties that we establish in this work hold on those benchmarks. Namely, domain connectivity phenomenon holds between all pairs of math-instruct metrics. For quality increase upon head rewind we obtain the following results: on MATH-500 rewind of domain-specific CPT 7B checkpoint results in an increase of 4.6% and for OLMo-2 CPT 7B checkpoint - 1%; on SQuAD rewind of domain-specific CPT 7B checkpoint yields an increase of 1.34% and for OLMo-2 CPT 7B checkpoint - 2.24%.
>
> [1]: Grattafiori, Aaron, et al. "The llama 3 herd of models." _arXiv preprint arXiv:2407.21783_ (2024).
>
> [2]: Yang, An, et al. “Qwen2.5 Technical Report.” arXiv, 2024, arXiv:2412.15115.
>
> [3]: Biderman, Stella, et al. "Pythia: A suite for analyzing large language models across training and scaling." _International conference on machine learning_. PMLR, 2023.
>
> [4]: Allal, Loubna Ben, et al. "SmolLM2: When Smol Goes Big--Data-Centric Training of a Small Language Model." _arXiv preprint arXiv:2502.02737_ (2025).
>
> [5]: Liu, Zhengzhong, et al. "Llm360: Towards fully transparent open-source llms." _arXiv preprint arXiv:2312.06550_ (2023).
>
> [6]: Zhang, Ge, et al. "Map-neo: Highly capable and transparent bilingual large language model series." _arXiv preprint arXiv:2405.19327_ (2024).
>
> [7]: Mo, Kaixiang, et al. "Mid-training of large language models: A survey." _arXiv preprint arXiv:2510.06826_ (2025).
>
> [8]: Abdin, Marah, et al. "Phi-4 technical report." _arXiv preprint arXiv:2412.08905_ (2024).
>
> [9]: Jordan, Keller et al. Muon: An optimizer for hidden layers in neural networks. 2024. URL: https://kellerjordan.github.io/posts/muon/
>
> [10]: Lightman, Hunter, et al. "Let's verify step by step." _The twelfth international conference on learning representations_. 2023.
>
> [11]: Rajpurkar, Pranav, et al. "Squad: 100,000+ questions for machine comprehension of text." _Proceedings of the 2016 conference on empirical methods in natural language processing_. 2016.

---

> > ### Author Rebuttal · Reviewer_YNp2 · 2026-04-01
> >
> > The additional benchmarks help to strengthen the claims and increase the trustworthiness of the work. I still have concerns about the lack of analysis on other LLMs and LLMs larger than 7B. I agree that the 7B/8B size is common across many model families, however, so are other sizes like 13B, 30B, and 70B. I sympathize with the concern about computational resources and time constraints in evaluating other models, but doing so would significantly strengthen the generalizability claims as well as increase the impact of this study. Without this expansion, the best option is to contextualize the claims to smaller LLMs (8B and below) in the abstract, discussion, and the new Limitations section. Having a limitations section certainly helps, but many readers will focus on the abstract so clarifications should be made there as well. Can you expand on your plan to move forward in addressing these concerns?

---

> > > ### Author Response · Authors · 2026-04-06
> > >
> > > We thank the reviewer for this constructive follow-up. We have carried out additional experiments on OLMo-2 models at 32B and 13B scales. We observe consistent quality improvements upon head rewind for both math and instruct domain CPT checkpoints, as well as for the original OLMo-2 checkpoints, and confirm the presence of domain connectivity for the 13B model. We will make sure to delineate the updated scope of our work in the "Limitations" section.
> > >
> > > For the 32B model - the largest available in the OLMo-2 suite - we apply head rewind to the publicly available CPT checkpoints released by the OLMo team and observe improvements of up to 3.6% on GSM8K and 1.2% on MATH-500, as well as up to 0.5% on DROP and 0.1% on SQuAD.
> > >
> > > For the 13B model, we perform math and instruct CPT runs with a 10B token budget. We note that the resulting checkpoints achieve quality close to the publicly available OLMo-2 13B CPT checkpoints trained on 100B tokens: the quality gap is 7% on DROP and only 0.2% on GSM8K. Interpolation between these domain-specific checkpoints has allowed us to confirm the presence of domain connectivity at this model scale.
> > >
> > > Moreover, we observe consistent quality improvements upon head rewind:
> > > 1. For the 13B math domain-specific CPT checkpoint, head rewind yields gains of 1.9% on GSM8K and 2.2% on MATH-500;
> > > 2. For the 13B instruct domain-specific CPT checkpoint, head rewind yields smaller gains of 0.1% on DROP and 0.3% on SQuAD;
> > > 3. For the released OLMo-2 13B CPT checkpoints, head rewind also yields improvements: 0.1% on both DROP and SQuAD, and 6.3% on GSM8K, 1.4% on MATH-500.
> > >
> > > In summary, the new experiments have allowed us to demonstrate consistent model quality increases upon head rewind, which we observe even at the 32B scale. We plan to further investigate its origins and the effect of head rewind on the downstream stages of the LLM training pipeline. We also intend to widen our analysis of the domain connectivity phenomenon. Finally, by releasing the _Diffract_ package, we aim to enable the research community to apply our analysis more broadly.
> > >
> > > We sincerely thank the reviewer for the constructive feedback; it has been invaluable in strengthening both the empirical scope and the clarity of our claims.

---

### Official Review · Reviewer_bvwQ · 2026-03-13

**Soundness:** 4
**Presentation:** 3
**Significance:** 3
**Originality:** 3
**Overall Recommendation:** 5
**Confidence:** 4

**Summary:**

This paper carefully studies the properties of continually pre-trained models by applying singular value decomposition (SVD) to weight matrices. Through extensive experiments across different training token budgets, data mixtures, and model sizes, the paper examines checkpoints via SVD and reveals several interesting findings:
(1) The singular value spectra of weight matrices evolve significantly during pre-training — the distribution initially develops a heavy-tailed power-law shape, and with further training beyond 100B tokens, additional complexity emerges in the form of multiple narrow peaks and outliers, after which the spectra largely stabilize.
(2) With longer pre-training, singular vector directions are updated substantially during CPT, giving rise to so-called head heterogeneity, where different attention heads exhibit domain-specific behavior. Based on these two findings, the paper proposes an effective head-pruning criterion that preserves model performance and even yields improvements on certain tasks.

In addition, the paper presents extensive experiments on linear interpolation between CPT checkpoints across different domains, showing that in larger pre-training settings, models trained directly on mixed datasets generally outperform their linearly interpolated counterparts.

**Compliance With Llm Reviewing Policy:**

Affirmed.

**Final Justification:**

I think this is a good paper.

**Key Questions For Authors:**

See weakness above.

**Limitations:**

Yes

**Strengths And Weaknesses:**

**Strengths**
- This is a technically solid paper with extensive experiments, interesting findings, and credible conclusions. I believe these kinds of findings can be valuable for the community to better understand LLMs.
- The paper is significant: it reveals interesting findings about head heterogeneity and proposes a corresponding effective head-rewinding criterion — converting this finding into something practically useful.
- This line of work on interpretability of LLMs is a promising direction to guide better model training, and the findings revealed in this paper are meaningful for this direction.

**Weaknesses**

I think this is a good paper and only have few minor suggestions regarding the presention.
- Figure 1 and Section 4.1 breaks the flow. They contain nice plots and interesting findings, I appreciate the extensive experiment for drawing this figure, but I don't think they have a strong connection to the main finding --- head heterogeneity. They should either be moved to the domain connectivity related section or the appendix.
- The layout of Figure 3 needs improvement. The axis labels are too small; increasing the font size or bolding the text would improve readability.

---

> ### Author Rebuttal · Authors · 2026-03-31
>
> We appreciate these suggestions, which will help make the presentation clearer. Thank you for your encouraging feedback. Please do not hesitate to reach out with any further questions.

---

> > ### Author Rebuttal · Reviewer_bvwQ · 2026-04-06
> >
> > thanks the author for the response, I will keep my positive score.

---

### Official Review · Reviewer_UXmf · 2026-03-13

**Soundness:** 3
**Presentation:** 3
**Significance:** 2
**Originality:** 3
**Overall Recommendation:** 4
**Confidence:** 3

**Summary:**

The paper investigates the continual pre-training (CPT) stage of Large Language Models across multiple specialized domains, including mathematics, instruction, code, and natural text. By using singular value decomposition and an open-source toolkit they developed called Diffract , the authors analyze how weight matrices evolve during domain adaptation. They outline several novel phenomena, such as "head heterogeneity" , the ability to prune up to 60% of attention heads in the CPT delta without notable quality loss , and "domain connectivity," which allows for linear interpolation between models adapted to different domains. Additionally, they demonstrate that selectively rewinding low-importance attention heads to their pre-trained state can actually improve benchmark accuracy by up to 4%.

**Compliance With Llm Reviewing Policy:**

Affirmed.

**Final Justification:**

The rebuttal addressed my concerns, and I raised my score accordingly

**Key Questions For Authors:**

The findings in section 4.5 Domain connectivity are very interesting. Could you elaborate on your hypothesis a bit more: “We hypothesize that the transition from concave to convex behavior may be related to the development of complex spectral structure in attention heads”?

**Limitations:**

yes

**Strengths And Weaknesses:**

I think it is an interesting read. The experiments are sound, comprehensive, and well-designed, evaluating both 1B and 7B models across different token budgets and domain mixtures.

And the introduction of the head-importance criterion and the resulting 4% quality increase upon partial head rewind is a strong, measurable practical contribution.


However, the paper is mainly observations. I feel like I don’t have a clear understanding of why these phenomena happen. It would be better to include more analysis or intuition on each discovery.

I also think any practical guidance on how to develop CPT strategies while leveraging these spectral insights would greatly strengthen the paper. For instance, the authors note that linear interpolation underperforms compared to CPT trained directly on dataset mixtures, but they do not propose a concrete alternative merging strategy.

In addition, the paper states that “the CPT stage of the LLM training pipeline is significantly less studied compared to the supervised fine-tuning (SFT) stage, where a rich set of phenomena has been documented—including linear mode connectivity (Frankle et al., 2020), task arithmetic (Ilharco et al., 2023), model soups (Wortsman et al., 2022), ability transfer (Yu et al., 2024) and low-rank subspace modification (Hu et al., 2022)”. While the paper does present many interesting findings from each aspect for CPT, there’s a lacking discussion of how they contrast against the SFT stage. It would be highly interesting to see exactly how CPT differs from SFT across these dimensions.

---

> ### Author Rebuttal · Authors · 2026-03-31
>
> Thank you for your constructive feedback. We address the questions and concerns below.
>
> **Q.1**
>
> Firstly, we note that during the CPT stage the singular spectra remain almost constant, and changes in the weight matrices can be largely attributed to the changes in the singular vectors. Those changes can be expressed as orthogonal transformations $R_{h,u}$ and $R_{h,v}$ acting on the left and right singular subspaces:
> $$
> W_h^{\text{CPT}} = U_h^{\text{CPT}} \Sigma_h^{\text{CPT}} (V_h^{\text{CPT}})^\top
> \approx
> U_h^{\text{pre}} R_{h,u} \Sigma_h^{\text{pre}} (V_h^{\text{pre}}R_{h,v})^\top
> $$
> This allows us to analyze the changes in weight matrices through the lens of singular vector agreements, as defined in Appendix A.3:
> $$
> A^u(W_h^{\text{CPT}},W_h^{\text{pre}}) = \lvert (U_h^{\text{CPT}})^\top U_h^{\text{pre}} \rvert,
> $$
>
> Analyses analogous to that depicted in Figure 2b have allowed us to make an empirical observation: upon changes to a weight matrix the vector agreement decreases most prominently in the region of the singular value spectrum with the highest density of singular values, and the changes also affect adjacent singular vectors in a peak of singular spectrum. For checkpoints with small pre-train budgets the spectral shape does not change appreciably compared to a broad single-peak Marchenko-Pastur random matrix spectrum. During the CPT stage a lot of singular vectors change simultaneously for such checkpoints. As the pre-train budget increases, the spectral structure in attention heads weight matrices develops sharp peaks. For such checkpoints the number of vectors in a single peak of singular spectrum undergoing change is much smaller.
>
> We hypothesize that this phenomenon has two effects. First, this can give rise to the appearance of domain-specific heads, as it is more plausible that limited domain-specific adaptations can be contained in a smaller-dimensional subspace. Secondly, smaller number of vectors undergoing change in non-domain specific heads can facilitate model stability under linear interpolation, as most of the singular vectors, including leading ones, remain stable, inhibiting the model quality degradation.
>
> We note that this is a preliminary hypothesis, and we will expand on this line of thought in future work. We can expand the discussion in the main text to include the points above, if it will be considered appropriate.
>
> We would also like to briefly address the weaknesses mentioned in the review.
>
> **W.1**
>
> We note that the observed behavior of quality increase of checkpoints after CPT upon head rewind can be directly used in practice. To that end, we carry out head-level rewind on OLMo 2 checkpoint after CPT provided by the model developers and we observe quality increase on GSM8K benchmark of 1.02%, on MATH-500 of 1%, on DROP of 1.31%, on SQuAD of 2.24%. This shows that head-level rewind can be considered for incorporation into standard LLM training pipelines. We leave the investigation of behavior of such checkpoints after a full training pipeline of SFT and RL to further research.
>
> **W.2**
>
> One of the objectives of our work was to infer if the phenomena observed for the SFT, namely: linear mode connectivity, low-rank weight update structure, and task arithmetic - are present for the CPT stage. We summarize the results below.
>
> **Linear mode connectivity**:  SFT checkpoints exhibit linear mode connectivity (Frankle et al., 2020) when fine‑tuning from a common pre‑trained checkpoint. For CPT we discover a similar effect which we coin "domain connectivity". However we note that the quality of the interpolation is highly dependent on the pre-train budget, and we hypothesize that it can be related to the complex spectral structure.
>
> **LoRA**: A substantial number of parameter-efficient fine-tuning methods has been proposed, including those explicitly utilizing low-rank structure of SFT parameter updates. Motivated by those findings we study the spectral structure and rank of CPT deltas, and find that for 7B models only about 50% of singular values can be removed without a decrease in quality, making the low-rank approximations to CPT unlikely to succeed. However, we find another surprising effect, namely, that the CPT delta is more sparse along attention heads; moreover, selective rewinding of some attention heads can lead to a quality increase of the model, which we demonstrate on both domain-specific CPT checkpoints and public OLMo 2 checkpoints.
>
> **Task Arithmetic** :  In SFT the weight updates can be interpreted as additive task vectors (Ilharco et al., 2023), enabling straightforward merging. We have tried setting up task vector arithmetic for CPT, but those experiments did not yield positive results. We hypothesize this is due to the fact that the CPT updates are of high rank and are not as compressible as SFT weight deltas.
>
> We will be happy to expand the discussion in the main text for improved clarity.

---

> > ### Author Rebuttal · Reviewer_UXmf · 2026-04-04
> >
> > Thanks for the responses! My concerns are fully addressed

---

### Decision · Program_Chairs · 2026-04-30

**Decision:**

Accept (spotlight)

**Comment:**

This paper proposes a new way of understanding LLM continual pre-training. It views adaptation as a low-dimensional directional rotation of weight singular vectors in a structured space. Most changes are around dominant spectral components. The singular spectrum itself, however, remains stable during training. The paper further shows that attention heads behave very differently during adaptation: some change across all domains, while others change only for specific domains. In other words, most of the attention head updates can be removed with little performance loss. Rewinding low-importance heads back to pre-trained values can improve accuracy. The paper also shows that models adapted to different domains can be linearly interpolated. The authors release Diffract, a scalable spectral analysis tool for large models.

The main reasons for acceptance are the following:
- Interesting and novel empirical findings on the continual pretraining dynamics. Empirical results are well-executed.
- Application of spectral analysis to continual pretraining and domain adaptation is novel.
- Practical contributions on attention head removement and rewinding.
- An open-source tool is useful for the community.

Some of the main concerns include the lack of deep theoretical results, limited to observational phenomena, and limited intuition for the why question, limited model families and benchmarks, and some presentation issues. Many of these issues have been clarified during the rebuttal. I recommend acceptance.